# PREDICTION WITHOUT PRECLUSION: RECOURSE VERIFICATION WITH REACHABLE SETS

**Avni Kothari**[†]
UCSF

**Bogdan Kulynych**[†]
EPFL

**Tsui-Wei Weng**
UCSD

**Berk Ustun**
UCSD

## ABSTRACT

Machine learning models are often used to decide who receives a loan, a job interview, or a public benefit. Models in such settings use features without considering their *actionability*. As a result, they can assign predictions that are *fixed* – meaning that individuals who are denied loans and interviews are, in fact, *precluded from access* to credit and employment. In this work, we introduce a procedure called *recourse verification* to test if a model assigns fixed predictions to its decision subjects. We propose a model-agnostic approach for recourse verification with *reachable sets* – i.e., the set of all points that a person can reach through their actions in feature space. We develop methods to construct reachable sets for discrete feature spaces, which can certify the responsiveness of *any model* by simply querying its predictions. We conduct a comprehensive empirical study on the infeasibility of recourse on datasets from consumer finance. Our results highlight how models can inadvertently preclude access by assigning fixed predictions and underscore the need to account for actionability in model development.

## 1 INTRODUCTION

Machine learning models routinely assign predictions to *people* – be it to approve an applicant for a loan [24], a job interview [5, 51], or a public benefit [66, 13, 16]. Models in such applications use features about individuals without accounting for how individuals can change them. In turn, they may assign predictions that are not *responsive* to the *actions* of their decision subjects. In effect, even the most accurate model can assign a prediction that is *fixed* (see Fig. 1).

The responsiveness of machine learning models to our actions is vital to their safety in consumer-facing applications. In applications like content moderation, models *should* assign fixed predictions to prevent malicious actors from circumventing detection by manipulating their features [25, 42, 31]. In lending and hiring, however, predictions should exhibit *some* sensitivity to our actions. Otherwise, models that deny loans and interviews may *preclude access* to credit and employment, thus violating basic rights such as equal opportunity [3] and universal access [8].

In this work, we introduce a formal verification procedure to test the responsiveness of a model's predictions with respect to the actions of its decision subjects. Our procedure – *recourse verification* – is grounded in a stream of work on algorithmic recourse [57, 59, 28]. While much of the work in this area focuses on *recourse provision* – i.e., providing a person with actions to obtain a desired prediction from a model – we focus on recourse verification – i.e., certifying that a model assigns predictions that each person can change. Unlike provision, verification is a model auditing procedure that practitioners can use to flag models that preclude access or promote manipulation.

The key challenge in recourse verification stems from the fact that we must test the sensitivity of a model's predictions with respect to actions rather than arbitrary changes in feature space. In a lending application, for example, actions on a feature such as `years_of_account_history` should set its value to a positive integer and should lead to a commensurate change in other temporal features like age. Such constraints are easy to specify for features that are semantically meaningful, but difficult to enforce in methods for recourse provision. To claim that a model assigns a fixed prediction to a point, we must prove that its predictions will not change under any possible action. In practice,

---

[†]Equal Contribution

**Empirical Risk Minimization**
most accurate classifier assigns
**prediction without recourse** to (1, 1)

| reapplicant | age $\geq 60$ | Dataset | | Best Model | |
|---|---|---|---|---|---|
| $x_1$ | $x_2$ | $n^-$ | $n^+$ | $\hat{f}$ | $\hat{R}(\hat{f})$ |
| 0 | 0 | 10 | 25 | + | 10 |
| 0 | 1 | 11 | 25 | + | 11 |
| 1 | 0 | 12 | 25 | + | 12 |
| 1 | 1 | 27 | 15 | − | 15 |

**Action Set**
actionability constraints that we can
elicit and embed into optimization problems

$$A(x_1, x_2) = \left\{ \begin{array}{c} a_1 \geq 0 \\ a_2 \geq 0 \\ a_1 + x_1 \in \{0,1\} \\ a_2 + x_2 \in \{0,1\} \end{array} \right\}$$

**Reachable Sets**
sets of feature vectors that allow for
**post-hoc** recourse verification for **any model**

$R_A(0,0) = \{(0,0),(0,1),(1,0),(1,1)\}$
$R_A(0,1) = \{(0,1),(1,1)\}$
$R_A(1,0) = \{(1,0),(1,1)\}$
$R_A(1,1) = \{(1,1)\}$

**Figure 1:** Stylized classification task where the most accurate classifier on a dataset with $n^- = 60$ negative examples and $n^+ = 90$ positive examples assigns a prediction without recourse to individuals with $(x_1, x_2) = (1, 1)$. We predict $y = \texttt{repay\_loan}$ using two binary features $(x_1, x_2) = (\texttt{reapplicant}, \texttt{age} \geq \texttt{60})$, which can only increase from 0 to 1. We denote the actions on each feature as $(a_1, a_2)$ and show the constraints they must obey in the *action set*. Given any model, we certify the responsiveness of its outputs for $(x_1, x_2)$ by checking its prediction for each point in the *reachable set* $R_A(x_1, x_2)$. In this case, 42 individuals with $(x_1, x_2) = (1, 1)$ are assigned a prediction without recourse as $f(x_1, x_2) = 0$ for all $(x_1, x_2) \in R_A(1, 1)$.

this requires an exhaustive search over a combinatorial subset of actionable feature space. This is a non-trivial computational task – especially for complex models – as we must certify the infeasibility of a combinatorial optimization problem that faithfully encodes a complex decision boundary.

Our main contributions include:

1. We present a model-agnostic approach for recourse verification by constructing a *reachable set* – i.e., a set of all points that a person can attain through their actions in feature space. Given a reachable set, we can certify the responsiveness of a model's predictions by simply querying its predictions over reachable points.

2. We develop fast methods to construct reachable sets for discrete feature spaces. Our methods can construct complete reachable sets for complex actionability constraints, and can support practical verification in model development and deployment.

3. We conduct a comprehensive empirical study on the infeasibility of recourse in consumer finance applications. Our results show how models can assign fixed predictions due to inherent actionability constraints, and demonstrate how existing methods to generate recourse actions and counterfactual explanations may inflict harm by failing to detect such instances.

4. We develop a Python package for recourse verification with reachable sets. Our package includes an API for practitioners to easily specify complex actionability constraints, and routines to test the actionability of recourse actions and counterfactual explanations.

**Related Work** We focus on a new direction for algorithmic recourse [57, 59, 29] – i.e., as a procedure to certify the responsiveness of a model's predictions with respect to the actions of its decision subjects. Although actionability is a defining characteristic of recourse [see 59], few works mention models may assign fixed predictions as a result of actionability constraints [57, 30, 10]. The lack of awareness stems, in part, from the fact that methods for recourse provision are typically designed and evaluated with simple actionability constraints such as immutability and monotonicity. As we show in Appendix C.5, however, infeasibility only arises once we start to consider actionability constraints that are difficult to handle in algorithm design.

We study recourse verification as a model auditing procedure to safeguard access in applications like lending. In such applications, verification is essential for reliable recourse provision – as it can flag a model that cannot provide recourse to consumers before it is deployed. To this end, our motivation aligns with a stream of work on the robustness of recourse provision with respect to distribution shifts [52, 18, 2, 47, 20], model updates [56, 49], and causal effects [40, 28, 35]. More broadly, recourse verification is a procedure to test the responsiveness of predictions over semantically meaningful features, which may be useful for stress testing for counterfactual invariance [58, 41, 50], certifying adversarial robustness on tabular datasets [39, 27, 23, 60, 42, 31], or designing models that incentivize improvement or deter strategic manipulation [19, 11, 38, 44, 15, 6, 21, 54, 32, 33, 1, 22].

## 2 RECOURSE VERIFICATION

We consider a standard classification task where we are given a model $f : \mathcal{X} \to \mathcal{Y}$ to predict a label $y \in \mathcal{Y} = \{0, 1\}$ from a vector of features $\boldsymbol{x} = [x_1, \dots, x_d] \in \mathcal{X}$ in a bounded feature space $\mathcal{X}$. We assume each instance represents a person, and that $f(\boldsymbol{x}) = 1$ represents a desirable *target prediction* – e.g., an applicant with features $\boldsymbol{x}$ will repay a loan within 2 years.

Our goal is to test if each person can obtain a target prediction from the model by changing their features. We represent such changes in terms of actions. Formally, each *action* as a vector $\boldsymbol{a} = [a_1, \dots, a_d] \in \mathbb{R}^d$ that shifts their features from $\boldsymbol{x}$ to $\boldsymbol{x} + \boldsymbol{a} = \boldsymbol{x}' \in \mathcal{X}$. We refer to the set of all actions from $\boldsymbol{x} \in \mathcal{X}$ as an *action set* $A(\boldsymbol{x})$, and assume that it contains a *null action* $\boldsymbol{0} \in A(\boldsymbol{x})$. In practice, an action set $A(\boldsymbol{x})$ is a collection of constraints. As shown in Table 1, we can express these constraints in natural language or as equations that we can embed into an optimization problem.

Semantically meaningful features admit hard actionability constraints. In the simplest cases, actionability constraints reflect the way that semantically meaningful features can only be altered in specific ways. For example, a model may use a feature that cannot change (e.g., age) or that can only be changed in specific ways (e.g., has_phd, which can only be changed from 0 to 1). More generally, constraints may require that changing one feature will induce changes on other features (e.g., changing married from 0 to 1 must set single from 1 to 0). Such downstream effects can be directional (e.g., changing retired from 1 to 0 will set work_days_per_week to 0, but not vice-versa), and may affect features that are not themselves actionable (e.g., changing years_of_account_history from 0 to 2 will increase age by 2 years).

| Class | Separable | Discrete | Example | Features | Constraint |
|---|---|---|---|---|---|
| Immutability | ✓ | ✗ | n_dependents should not change | $x_j = $ n_dependents | $a_j = 0$ |
| Monotonicity | ✓ | ✗ | reapplicant can only increase | $x_j = $ reapplicant | $a_j \geq 0$ |
| Integrality | ✓ | ✓ | n_accounts must be positive integer $\leq 10$ | $x_j = $ n_accounts | $a_j \in \mathbb{Z} \cap [0 - x_j, 10 - x_j]$ |
| Categorical Encoding | ✗ | ✓ | preserve one-hot encoding of married, single | $x_j = $ married $x_k = $ single | $a_j + x_l \in \{0,1\} \quad x_k + a_k \in \{0,1\}$ $a_j + x_j + a_k + x_k = 1$ |
| Ordinal Encoding | ✗ | ✓ | preserve one-hot encoding of max_degree_BS, max_degree_MS | $x_j = $ max_degree_BS $x_k = $ max_degree_MS | $a_j + x_j \in \{0,1\} \quad x_k + a_k \in \{0,1\}$ $a_j + x_j + a_k + x_k = 1 \quad a_j + x_j \geq a_k + x_k$ |
| Logical Implications | ✗ | ✓ | if is_employed = TRUE then work_hrs_per_week $\geq 0$ else work_hrs_per_week $= 0$ | $x_j = $ is_employed $x_k = $ work_hrs_per_week | $a_j + x_j \in \{0,1\}$ $a_k + x_k \in [0, 168]$ $a_j + x_j \leq 168(x_k + a_k)$ |
| Causal Implications | ✗ | ✓ | if years_of_account_history increases then age will increase commensurately | $x_j = $ years_at_residence $x_k = $ age | $a_j \leq a_k$ |

**Table 1:** Examples of deterministic actionability constraints. We show how each constraint can be expressed in natural language and embedded into an optimization problem using standard techniques in mathematical programming [see e.g., 65]. We highlight constraints that are discrete and non-separable because they can only be enforced using special kinds of search algorithms.

**Verification as a Feasibility Problem** Given a model $f : \mathcal{X} \to \mathcal{Y}$ and a point $\boldsymbol{x} \in \mathcal{X}$ with action set $A(\boldsymbol{x})$, the *recourse provision* task seeks to find an action $\boldsymbol{a} \in A(\boldsymbol{x})$ that minimizes a cost function $\text{cost}(\boldsymbol{a} \mid \boldsymbol{x})$. This task requires finding an optimal solution to an optimization problem as in Eq. (1).

The *recourse verification* task seeks to determine if recourse is infeasible from $\boldsymbol{x}$ – i.e., if a model assigns the same prediction $f(\boldsymbol{x} + \boldsymbol{a}) = 0$ for all actions $\boldsymbol{a} \in A(\boldsymbol{x})$. This task only requires finding a feasible solution to Eq. (1), which can be cast as the optimization problem in Eq. (2).[1]

$$
\begin{array}{ll}
\text{Recourse Provision} & \\
\min & \text{cost}(\boldsymbol{a} \mid \boldsymbol{x}) \\
\text{s.t.} & f(\boldsymbol{x} + \boldsymbol{a}) = 1 \qquad (1) \\
& \boldsymbol{a} \in A(\boldsymbol{x})
\end{array}
\qquad
\begin{array}{ll}
\text{Recourse Verification} & \\
\min & 1 \\
\text{s.t.} & f(\boldsymbol{x} + \boldsymbol{a}) = 1 \qquad (2) \\
& \boldsymbol{a} \in A(\boldsymbol{x})
\end{array}
$$

---

[1]We set the objective of Eq. (2) to a constant so that any algorithm that solves Eq. (2) will terminate as soon as it has found a feasible solution.

We can write the input and output of a recourse verification method as the function:

$$\text{Recourse}(\boldsymbol{x}, f, A) = \begin{cases} \text{Yes}, & \text{if method returns an action } \boldsymbol{a} \in A(\boldsymbol{x}) \text{ such that } f(\boldsymbol{x} + \boldsymbol{a}) = 1 \\ \text{No}, & \text{if method proves that } f(\boldsymbol{x} + \boldsymbol{a}) = 0 \text{ for all actions } \boldsymbol{a} \in A(\boldsymbol{x}) \\ \bot, & \text{otherwise} \end{cases}$$

We say that a method for recourse verification *certifies feasibility* from $\boldsymbol{x}$ if it outputs Yes and that it *certifies infeasibility* from $\boldsymbol{x}$ if it outputs No. In practice, existing methods for recourse provision may return outputs that cannot support either of these claims. For example, they may fail to return an action without having searched exhaustively, or return an "action" that violates actionability constraints. In such cases, we say that the method *abstains* for $\boldsymbol{x}$ and denote its output as $\bot$.

**Use Cases**  Recourse verification is a model auditing procedure to test the *responsiveness* of a model's predictions with respect to the actions of its decision subjects. We can apply this procedure to flag models that are unsafe in different consumer-facing applications by testing responsiveness by choosing an appropriate action set.

*Detecting Preclusion.* In applications where we would like to safeguard access (e.g., lending), we can flag that a model $f$ precludes access by testing the responsiveness of predictions on points for which $f(\boldsymbol{x}) = 0$. In this case, we would specify an action set that captures indisputable constraints and applies to all individuals. We would claim that the model precludes access if $\text{Recourse}(\boldsymbol{x}, f, A) = \text{No}$ for any point such that $f(\boldsymbol{x}) = 0$.

*Ensuring Robustness.* In applications where we would like to mitigate gaming (e.g., content moderation), we can certify that a model $f$ is vulnerable to adversarial manipulation by testing the responsiveness of its predictions on points for which $f(\boldsymbol{x}) = 0$. In this case, we would specify an action set $A(\boldsymbol{x})$ that encodes a threat model [31] – i.e., actions that let individuals obtain a target prediction by changing spurious features [see 15, 43]. We would claim that the model is vulnerable to manipulation if $\text{Recourse}(\boldsymbol{x}, f, A) = \text{Yes}$ for any point such that $f(\boldsymbol{x}) = 0$.

Since these audits apply over points in feature space, we can run verification at different stages of a model lifecycle to minimize the chances of inflicting harm. In model development, we would test if a model assigns fixed predictions to any point in the training data. In deployment, we would repeat this test for new points. In both cases, the procedure would establish that a model assigns fixed predictions, and could support further interventions to mitigate these effects (see Section 4).

Actionability can vary substantially between individuals [see 4, 59]. In principle, we can account for these variations by calling a recourse verification method with *personalized actionability constraints* that we elicit from each decision subject [via, e.g., an interface as in 62]. In practice, we can mitigate harm in consumer-facing applications without eliciting personalized constraints. This is because models may assign fixed predictions as a result of *inherent actionability constraints* – i.e., constraints that apply to all decision subjects and that practitioners could glean from a data dictionary (e.g., constraints that enforce physical limits or preserve a feature encoding). Seeing how inherent constraints represent a subset of personalized constraints, audits with inherent actionability constraints should be used to flag that a model inflicts harm rather than to certify that it is safe.

**Algorithm Design Requirements and Pitfalls**  Methods for recourse verification should be designed to *certify infeasibility*. This is an essential requirement for verification – as it implies that a method can prove that a model's prediction will not change under any possible action. The vast majority of existing methods for recourse provision are ill-suited for verification because cannot certify infeasibility. In practice, these methods will return outputs that are inconclusive or incorrect for recourse verification tasks. We refer to these instances as *loopholes* and *blindspots* and define them below.

**Definition 1.**  Given a recourse verification task for a model $f$ for a point $\boldsymbol{x}$ with the action set $A(\boldsymbol{x})$, we say that a method returns a *loophole* if its output violates actionability constraints.

Methods for recourse provision return loopholes when they search for actions using an algorithm that cannot enforce all actionability constraints in a recourse verification task. For example, methods that search for recourse actions using gradient descent [45] will return loopholes when we must verify recourse with respect to an action set that includes the discrete actionability constraints in Table 1.

**Definition 2.** Given a recourse verification task for a model $f$ for a point $x$ with the action set $A(x)$, we say that a method exhibits a *blindspot* if it fails to find an action for a point where $\text{Recourse}(x, f, A) = \text{Yes}$.

Methods for recourse provision output blindspots when they cannot search exhaustively. Common algorithm design patterns that lead to blindspots include: (i) Searching for actions over observed data points [see e.g., 61, 46]; and enforcing actionability by post-hoc *filtering* – i.e., by generating a large collection of changes in feature space and filtering them to enforce actionability [see e.g., 37] – which exhibits blindspots when the generation step is guaranteed to generate all possible actions.

## 3    VERIFICATION WITH REACHABLE SETS

We introduce a model-agnostic approach for recourse verification. Our approach constructs *reachable sets* – i.e., sets of feature vectors that obey actionability constraints.

**Definition 3.** Given a point $x$ and its action set $A(x)$, a *reachable set* contains all feature vectors that can be attained using the actions in $A(x)$: $R_A(x) := \{x + a \mid a \in A(x)\}$.

Given a reachable set $R_A(x)$, we can certify that a model $f$ provides recourse to by querying its predictions on each point $x \in R_A(x)$. Thus, we can write the verification function as:

$$\text{Recourse}(x, f, R) = \begin{cases} \text{Yes}, & \text{if there exists a reachable point } x' \in R_A(x) \text{ s.t. } f(x') = 1 \\ \text{No}, & \text{if } f(x') = 0 \text{ for all reachable points } x' \in R_A(x) \\ \perp, & \text{if } f(x') = 0 \text{ for some reachable points } x' \in R \subset R_A(x) \end{cases} \quad (3)$$

Verification with reachable sets has three key benefits:

*Model Agnostic Verification*: We can use reachable sets to verify recourse for *any model class*. Model agnostic approaches are especially valuable for recourse verification because it is challenging, if not impossible, to develop a model-specific approach for complex model classes such as ensembles and deep neural networks. In particular, this stems from the fact that such method would have to certify the infeasibility of a combinatorial optimization problem that encodes both the model and the actionability constraints. In practice, such problems be prohibitively large to solve in an audit – as we would have to encode a complex decision boundary [see e.g., 55, 48].

*Amortization*: In a recourse verification task where we have access to a suitable method for recourse verification, we may still wish to verify recourse using a reachable set. This is because, once we have constructed reachable sets, we can use them to verify recourse for as many models as we wish.

*Explicit Abstention*: In settings where we cannot enumerate a complete reachable set, we can use the interior approximation of the reachable set $R \subset R_A(x)$. In this case, the procedure will certify recourse if it can find a feasible action. Otherwise, it will abstain – thus, flagging $x$ as a *potential* prediction without recourse. We can exploit this property to speed up construction through a lazy initialization pattern. For example, rather than constructing a complete reachable set for every training example, we can construct an interior approximation $R \subset R_A(x)$. In this setup, we would use the interior approximations to certify feasibility, and only construct the full reachable sets $R = R_A(x)$ for points for which we would abstain $\text{Recourse}(x, f, R) = \perp$.

### 3.1    CONSTRUCTION

In Algorithm 1, we present a procedure to construct a reachable set for a given point by solving an optimization problem of the form:

$$\text{FindAction}(x, A) := \underset{}{\arg\min} \ \|a\| \ \text{ s.t. } \ a \in A(x) \setminus \{0\}.$$

We formulate $\text{FindAction}(x, A)$ as a mixed-integer program that we present in Appendix B. Our formulation can encode all actionability constraints in Table 1 and is designed to be solved in a way that is fast and reliable using an off-the-shelf solver [see e.g., 17, for a list].

| **Algorithm 1** GetReachableSet |
|---|
| **Require:** $x \in \mathcal{X}$, feature vector |
| **Require:** $A(x)$, action set for $x$ |
| $\quad R \leftarrow \{x\}$ |
| $\quad A \leftarrow A(x)$ |
| 1: **while** $\text{FindAction}(x, A)$ is feasible **do** |
| 2: $\quad a^* \leftarrow \text{FindAction}(x, A)$ |
| 3: $\quad R \leftarrow R \cup \{x + a^*\}$ |
| 4: $\quad A \leftarrow A \setminus \{a^*\}$ |
| **Output** $R = R_A(x)$ |

Given a point $\boldsymbol{x}$, the procedure enumerates all reachable points by repeatedly solving this problem and removing prior solutions by adding a "no-good" constraint [see e.g., 53]. The procedure stops once $\mathsf{FindAction}(\boldsymbol{x}, A)$ is infeasible – at which point it has enumerated all possible actions and thus reachable points. In practice, the procedure can be stopped when a user-specified stopping condition is met, in which case it would return an interior reachable set $R \subset R_A(\boldsymbol{x})$ that can certify feasibility.

**Decomposition**   Seeing how reachable sets grow exponentially with the number of features, Algorithm 1 may generate an incomplete reachable set that cannot certify infeasibility under reasonable time constraints. We overcome this issue through a decomposition – i.e., by applying Algorithm 1 to subsets of features that can be altered independently for all points $\boldsymbol{x} \in \mathcal{X}$.[2]

Given an action set over $d$ features, we can identify subsets that can be altered independently by inspection. In this way, we can construct the *most granular partition* of features – i.e., a collection of $k \leq d$ feature subsets $\mathcal{M} := \{S_1, \ldots, S_k\}$ such that $A(\boldsymbol{x}) = \prod_{S \in \mathcal{M}} A_S(\boldsymbol{x}_S)$. Given the partition $\mathcal{M}$, we generate reachable sets for each feature subset $R_S$ by calling Algorithm 1 for each $A_S(\boldsymbol{x}_S)$, and recover the full reachable set as $R = \prod_{S \in \mathcal{M}} R_S$.

Decomposition moderates the combinatorial explosion in our setting – making it viable to enumerate reachable sets in practice. This strategy leads to considerable improvement in runtime, as we construct reachable sets for each subset by solving smaller instances of $\mathsf{FindAction}()$, and can construct the reachable set for a single feature subsets without solving a MIP.

## 3.2   AUDITING IN PRACTICE

We can verify recourse in model development by constructing a reachable set for each point in a dataset. Once we have constructed reachable sets for each point, we can call recourse verification for any model by querying its predictions on reachable points as per Eq. (3). In practice, the most time-consuming part of our approach stems from the construction of reachable sets. In our implementation, we can achieve a considerable speed up in construction through parallel computing and sharing reachable sets across points. In a task with immutable features, for example, we only need to construct and store a single reachable set for any points $\boldsymbol{x}$ and $\boldsymbol{x}'$ that only differ in terms of immutable feature values.

Given that our approach is designed to verify recourse with prediction queries, it may be time-consuming for models with a resource-intensive inference step. In such cases, we can minimize prediction queries through short-circuiting. In some settings, we can certify that a model provides recourse to a point analytically – i.e., without querying its predictions on reachable points – by applying the result in Theorem 4.

**Theorem 4.** *Suppose we have a dataset $\mathcal{D} = \{(\boldsymbol{x}_i, y_i)\}_{i=1}^n$ with $n^+$ positive examples, and a point $\boldsymbol{x}$ with the reachable set $R \subseteq R_A(\boldsymbol{x})$. In this case, every model $f : \mathcal{X} \to \mathcal{Y}$ will provide recourse to $\boldsymbol{x}$ so long as its false negative rate over $\mathcal{D}$ obeys:*

$$\mathsf{FNR}(f) < \frac{1}{n^+} \sum_{i=1}^n \mathbb{1}[\boldsymbol{x}_i \in R \ \wedge \ y_i = 1]$$

Theorem 4 highlights an alternative approach for recourse verification with reachable sets – i.e., we can certify that a model $f$ must provide recourse to a point $x$ so long as the false negative rate does not exceed the density of positive examples in its reachable set. The values can be computed on *any dataset* with labels – be it the training dataset or a separate dataset. In practice, this approach may be useful when working with model classes where prediction queries are time-consuming. In practice, the result requires a dataset that is "dense" enough so that a reachable set for a point contains other labeled examples. When this condition holds, we can certify that a model provides recourse to a point by comparing the false negative rate of $f$ to the prevalence of positive examples in its reachable set.

---

[2]Formally, we say two subsets of features $S, T \subseteq [d]$ can be altered independently if the action set over $S \cup T$ can be expressed as a product of action sets over $S$ and $T$ for all points $\boldsymbol{x} \in \mathcal{X}$. For example, given the subsets $S, T$ where $S \cup T = [d]$, we write $A(\boldsymbol{x}) = A_S(\boldsymbol{x}_S) \times A_T(\boldsymbol{x}_T)$ for all $\boldsymbol{x} = [\boldsymbol{x}_S, \boldsymbol{x}_T] \in \mathcal{X}$ where $\times$ denotes a Cartesian product.

## 3.3 DISCUSSION AND EXTENSIONS

Our methods are designed to construct reachable sets that can be used for recourse verification over discrete feature spaces. In principle, we can construct reachable sets for continuous feature spaces through sampling, but leave this as a topic for future work as it involves a probabilistic guarantee of infeasibility (see Section 5).

Our methods may be useful as a tool to enforce actionability over continuous feature spaces. In particular, we can extend our formulation for FindAction() as a routine to test the feasibility of changes from existing methods to generate recourse actions and counterfactual explanations. In a case where such methods would suggest that a person can change their prediction by altering their features from $x$ to $x'$, we can test the feasibility such changes with respect to actionability constraints by solving an optimization problem of the form:

$$\text{IsReachable}(x, x', A) \quad := \quad \min \ 1 \quad \text{s.t.} \quad x = x' - a, \ a \in A(x). \tag{4}$$

This routine can be used as a way to test for actionability in existing methods via post-hoc filtering. In such cases, the resulting procedure would allow practitioners to flag outputs that violate actionability constraints, and avoid the challenges of detecting loopholes. As we explain in Section 2, it would not be able to certify that recourse is infeasible.

## 4 EXPERIMENTS

We present experiments showing how predictions without recourse arise under inherent actionability constraints and how existing methods can fail to detect these instances.

### 4.1 SETUP

We work with three classification datasets from consumer finance, where models that assign fixed predictions would preclude credit access (see Table 2). We process each dataset by encoding categorical attributes and discretizing continuous features. We use the processed dataset to fit a classification model using one of the following model classes: *logistic regression* (LR), *XGBoost* (XGB), and *random forests* (RF). We train each model using an 80%/20% train/test split and tune hyperparameters using standard $k$-CV. We report the performance of each model in Appendix C.

We specify inherent actionability constraints for each dataset – focusing on identifying indisputable conditions that apply to all individuals (e.g., compliance with physical limits, preserving feature encoding, enforcing deterministic causal effects, and preventing changes to protected attributes). We list the constraints for each dataset in Appendix C. We note that the constraints for all datasets include a mix of separable constraints (e.g. immutability, integrality, monotonicity) as well as non-separable constraints (e.g., encoding presentation, deterministic causal effects).

We construct reachable sets for each point in the dataset using Algorithm 1. We use the reachable sets to identify individuals who are assigned a prediction without recourse by any one of the models. The results from reachable sets reflect the ground-truth feasibility of recourse for each point and each model class. We label our results as Reach and use them to benchmark the reliability of two salient methods to generate recourse actions and counterfactual explanations:

- AR [57], a *model-specific* method that can certify infeasibility for linear classifiers and handle separable actionability constraints,
- DiCE [45], a *model-agnostic* method that handles some separable actionability constraints.

### 4.2 RESULTS AND DISCUSSION

**On Predictions without Recourse**  We summarize our results for each dataset, method, and model class in Table 2. Our results show that models assign fixed predictions under inherent actionability constraints. In practice, individuals who are assigned predictions without recourse may vary drastically across models that perform equally well. Seeing how reachable sets do not change across models, these differences arise from the different decision boundaries of each model.

| Dataset | Metrics | LR | | | XGB | | | RF | | |
|---|---|---|---|---|---|---|---|---|---|---|
| | | Reach | AR | DiCE | Reach | AR | DiCE | Reach | AR | DiCE |
| heloc $n = 5{,}842$ $d = 43$ FICO [14] | Certifies No Recourse | 22.2% | — | — | 22.3% | | — | 31.3% | | — |
| | Outputs Action | 77.8% | 85.9% | 57.6% | 77.7% | | 57.3% | 68.7% | | 49.3% |
| | ↳ Loopholes | **0.0%** | 41.1% | 34.4% | **0.0%** | NA | 42.1% | **0.0%** | NA | 29.5% |
| | Outputs No Action | 22.2% | 14.1% | 42.4% | 22.3% | | 42.7% | 31.3% | | 50.7% |
| | ↳ Blindspots | **0.0%** | 0.0% | 21.0% | **0.0%** | | 21.1% | **0.0%** | | 19.8% |
| german $n = 1{,}000$ $d = 36$ Dua and Graff [12] | Certifies No Recourse | 7.4% | — | — | 7.1% | | — | 28.6% | | — |
| | Outputs Action | 92.6% | 91.7% | 92.1% | 92.9% | | 93.3% | 71.4% | | 68.0% |
| | ↳ Loopholes | **0.0%** | 2.2% | 16.6% | **0.0%** | NA | 23.1% | **0.0%** | NA | 24.0% |
| | Outputs No Action | 7.4% | 8.3% | 7.9% | 7.1% | | 6.7% | 28.6% | | 32.0% |
| | ↳ Blindspots | **0.0%** | 1.3% | 0.9% | 0.0% | | 0.0% | **0.0%** | | 3.4% |
| givemecredit $n = 120{,}268$ $d = 23$ Kaggle [26] | Certifies No Recourse | 15.6% | — | — | 16.5% | | — | 0.2% | | — |
| | Outputs Action | 84.4% | 84.4% | 79.7% | 83.5% | | 78.5% | 99.8% | | 97.7% |
| | ↳ Loopholes | **0.0%** | 40.7% | 34.6% | **0.0%** | NA | 34.7% | **0.0%** | NA | 57.7% |
| | Outputs No Action | 15.6% | 15.6% | 20.3% | 16.5% | | 21.5% | 0.2% | | 2.3% |
| | ↳ Blindspots | **0.0%** | 0.0% | 4.7% | **0.0%** | | 5.0% | **0.0%** | | 2.3% |

**Table 2:** Overview of results for all datasets, model classes, and methods. For each dataset and model class, we use Reach to determine individuals who are assigned predictions without recourse. We use these results to reliability of AR and DiCE for recourse verification tasks. We evaluate each method in terms of the percentage of points where it: *certifies no recourse*; *outputs an action*; outputs a *loophole*, i.e., an action that violates actionability constraints; *outputs no action*; exhibits a *blindspot*, i.e., outputs no action when recourse exists. Here, each metric is expressed as a percentage of the points that are assigned a negative prediction by a model.

**On Loopholes** Our results in Table 2 show how methods to generate recourse actions may output *loopholes* – i.e., actions that violate actionability constraints. In particular, this failure mode affects between 2.2% to 57.7% of individuals across datasets, methods, and models. As we describe in Section 2, methods return loopholes when they cannot enforce *all* actionability constraints in a prediction task. In this case, we note that AR and DiCE can enforce separable actionability constraints. Thus, the loopholes arise from constraints that affect multiple features.

Loopholes reflect silent failures that undermine the benefits of recourse provision and may inflict harm. Consider a consumer finance application where we use AR or DiCE to provide consumers with actions that they can perform to qualify for a loan. In this case, loopholes that are left undetected would lead us to present consumers with recourse actions that are fundamentally impossible. On the heloc dataset, for example, we find that DiCE returns a loophole for 42.1% of individuals who are denied credit by an XGB model. Although some loopholes may be easy to spot through visual inspection or a basic immutability check, this is not always the case. In this case, $\approx 27\%$ of individuals receive an action that alters 5 or more features simultaneously – many of them can only be detected reliably through a programmatic approach that can test if they meet actionability constraints.

**On the Illusion of Feasibility** Our experiments show that recourse often *appears* to be feasible when methods are only able to enforce simple constraints. We study this effect in Appendix C through an ablation study where we audit models for the heloc dataset under special classes of actionability constraints. Our results show that methods return loopholes for individuals with fixed predictions under simple constraints such as immutability and monotonicity, and that infeasibility only arises once methods can enforce more complex constraints. In this case, we find that LR assigns a prediction without recourse to 22.2% of individuals. If we enforce monotonicity and integrality constraints, however, recourse *appears* to be feasible for $\approx 99\%$ points when using AR.

**On Blindspots** Our results show how existing methods for recourse provision may return results that are inconclusive or incorrect for verification. In Table 2, we highlight this failure mode by reporting the prevalence of *blindspots* – i.e., the proportion of instances where a method fails to return a recourse action for an individual who has recourse. On the heloc dataset, for example, we find that DiCE fails to find an action for 42.7% of individuals who are denied by the XGB model. In this case, DiCE returns an error message "no counterfactuals found for the given configuration, perhaps try with different parameters..." Our analysis shows that nearly half of these cases (21.1% of 42.7%) correspond to blindspots while the other half are predictions without recourse (21.6%).

Blindspots differ from loopholes in that they represent an "overt" failure mode that is unlikely to inflict harm. In practice, these failures are more likely to stump practitioners who find that a method

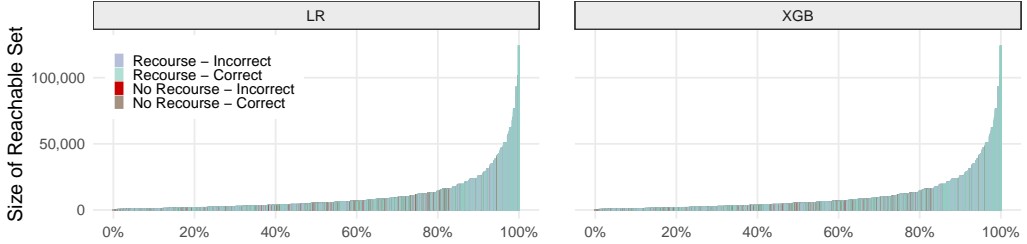

**Figure 3:** Reachable sets for each point in the `heloc` dataset, ordered from smallest to largest along the $x$-axis. We show predictions without recourse in red and highlight incorrectly predicted points in darker colors. As shown, predictions without recourse are prevalent across all reachable set sizes and can vary significantly between classifiers.

fails to find a recourse action. In such cases, the key issue is attribution, as practitioners cannot determine whether the failure is due to (1) a prediction without recourse; (2) the search algorithm used to search for actions; (3) a bug in the recourse provision package; (4) a bug in their code. More broadly, the results highlight the value of designing methods that can certify infeasibility – as methods that could certify infeasibility could provide evidence of preclusion in such cases.

**On Interventions to Mitigate Preclusion**   Our results show how recourse verification can guide heuristic interventions that mitigate preclusion. At a minimum, we can use the results from an audit for model selection – i.e., to choose a model that minimizes preclusion among models that are almost equally accurate. On the `heloc` dataset, for example, we find that RF and XGB models have a test AUC $\approx 0.780$ but an 11% difference in the predictions without recourse – thus, we can reduce preclusion without compromising performance by simply choosing to deploy an XGB model over an RF model. Seeing how reachable sets measure preclusion through prediction queries, we can apply this strategy in earlier stages of model development – e.g., we can search for model hyperparameters that minimize preclusion by defining a custom metric to compute the "preclusion rate." In this case, we note that parameters that control the decision boundary of the model can lead to substantial differences in preclusion without compromising training accuracy – as we only need to assign a target prediction to a reachable point rather than the current point. In general, we can mitigate preclusion by defining features to promote actionability or by dropping features that lead to fixed predictions. In contrast to the previous interventions, this may require constructing a new collection of reachable sets at each iteration.

## 5   CONCLUDING REMARKS AND LIMITATIONS

Our paper highlights how machine learning models can assign fixed predictions as a result of actionability constraints, and describes how such predictions can lead to preclusion in consumer-facing applications such as lending and hiring. Our work proposes to address these failure modes by developing methods for a task called recourse verification.

Recourse verification broadly represents a new direction for research in algorithmic recourse – i.e., as a model auditing procedure to certify the responsiveness of predictions with respect to actions. The methods in this paper are designed for recourse verification over discrete feature spaces and deterministic actionability constraints, but should be extended to address the following limitations:

- Our methods are designed to certify infeasibility with respect to actions over discrete feature spaces – and cannot certify infeasibility with respect to actions on continuous features. In principle, it is possible to construct reachable sets that certify infeasibility over continuous feature spaces by sampling. We leave this topic for future work as it requires a different algorithm and returns a probabilistic guarantee of infeasibility.
- Our methods do not consider probabilistic causal effects – i.e., where actions on a feature *may* incite changes on downstream features in a probabilistic causal model [see, e.g., 30, 34, 10, 35]. Although our methods may be useful to generate actionable interventions in this setting, a reliable method for verification should return a probabilistic guarantee of infeasibility that accounts for potential misspecification in the causal model.

ETHICS STATEMENT

Our work highlights how machine learning models can assign fixed predictions as a result of actionability, and proposes a task called recourse verification to reliably detect such instances.

We study recourse verification as a model auditing procedure that practitioners and auditors can use to detect preclusion in consumer-facing applications such as lending and hiring. The normative basis for the right to access in such applications stems from principles such as equality of opportunity (e.g., in hiring) and universal access (e.g., for basic health insurance). In other words, these are applications where we would want to safeguard access – even if it comes at a cost – because it reflects the kind of society we want to build. In practice, ensuring access may not impose any cost. In lending, for example, lenders only collect labels for consumers who are approved for loans [36, 9, 63, due to selective labeling]. Thus, consumers assigned predictions without recourse cannot generate labels that would signal creditworthiness [7]. In the United States, such effects have cut off credit access for large consumer segments whose creditworthiness is unknown [64, see, e.g., 26M "credit invisibles"].

Our paper primarily studies auditing models in lending and hiring with respect to indisputable constraints that apply to all decision subjects – *inherent actionability constraints*. Our recommendation is based on the fact that such constraints can be gleaned from a data dictionary, that claims surrounding preclusion should be indisputable, and that models may lead to preclusion as a result of such constraints. Given that individuals in these applications will face additional actionability constraints, the results of such an audit should be used to flag models that preclude access rather than to certify that models are safe.

In applications where elicitation is possible, our proposed approach can support a number of practices to handle assumptions surrounding actionability in a way that promotes transparency, contestability, and participatory design. In particular, individuals can express their constraints in natural language – allowing stakeholders to scrutinize and contest them even without technical expertise in machine learning. In the event that stakeholders disagree on actionability constraints, we recommend determining if their disagreements affect claims of infeasibility through an ablation study. In such cases, we can run verification using the subset of "consensus constraints" that all stakeholders agree on. In this worst case, we may still find that models lead to preclusion since the "consensus constraints" will always contain inherent constraints.

ACKNOWLEDGMENTS

This work is supported by the National Science Foundation (NSF) under grant IIS-2313105.

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

## A  PROOF OF THEOREM 4

**Theorem 4.** *Suppose we have a dataset of labeled examples* $\mathcal{D} = \{(\boldsymbol{x}_i, y_i)\}_{i=1}^n$. *Every model* $f : \mathcal{X} \to \mathcal{Y}$ *can provide recourse to* $\boldsymbol{x}$ *if:*

$$\mathsf{FNR}(f) < \frac{1}{n^+} \sum_{i=1}^n \mathbb{1}[\boldsymbol{x}_i \in R \ \wedge \ y_i = +1] \tag{5}$$

*where* $\mathsf{FNR}(f) := \frac{1}{n^+} \sum_{i=1}^n \mathbb{1}[f(\boldsymbol{x}_i) = -1 \ \wedge \ y_i = +1]$ *is the false negative rate of* $f$ *on* $\mathcal{D}$ *and where* $n^+$ *is number of positive examples in* $\mathcal{D}$, *and* $R \subseteq R_A(\boldsymbol{x})$ *is any subset of the reachable set.*

*Proof.* The proof is based on an application of the pigeonhole principle over the set of positive examples $S^+ := \{\boldsymbol{x}_i \mid y_i = +1, i \in [n]\}$. Given a classifier $f$, denote the number of true positive and false negative predictions over $S^+$ as:

$$\mathsf{TP}(f) := \sum_{i=1}^n \mathbb{1}[f(\boldsymbol{x}_i) = +1 \wedge y_i = +1] \qquad \mathsf{FN}(f) := \sum_{i=1}^n \mathbb{1}[f(\boldsymbol{x}_i) = -1 \wedge y_i = +1].$$

Consider a region of feature space $R \subseteq R_A(\boldsymbol{x})$ for which the number of correct positive predictions exceeds the number of positive examples outside $R$ so that:

$$\mathsf{TP}(f) > n^+ - |S^+ \cap R|.$$

In this case, the pigeonhole principle ensures that the classifier $f$ must assign a correct prediction to at least one of the positive examples in $R$ – i.e., there exists a point $\boldsymbol{x}' \in S^+ \cap R$ such that $f(\boldsymbol{x}') = y_i = +1$. Given $R \subseteq R_A(\boldsymbol{x})$, we have that $\boldsymbol{x} \in R_A(\boldsymbol{x})$. Thus, we can reach $\boldsymbol{x}'$ from $\boldsymbol{x}$ by performing the action $\boldsymbol{a} = \boldsymbol{x}' - \boldsymbol{x}$ – i.e., we can change the prediction from $f(\boldsymbol{x}) = -1$ to $f(\boldsymbol{x} + \boldsymbol{a}) = +1$.

We recover the condition in the statement of the Theorem as follows:

$$\mathsf{TP}(f) > n^+ - |S^+ \cap R| \tag{6}$$

$$\mathsf{FN}(f) < |S^+ \cap R|, \tag{7}$$

$$\mathsf{FNR}(f) < \frac{1}{n^+} \sum_{i=1}^n \mathbb{1}[\boldsymbol{x}_i \in R \ \wedge \ y_i = +1] \tag{8}$$

Here, we Eqn. (7) uses the fact that $\mathsf{TP}(f) = n^+ - \mathsf{FN}(f)$, and (8) divides both sides by $\frac{1}{n^+}$. The result follows by applying the definition of the false negative rate. $\qquad\square$

## B    REACHABLE SET GENERATION

In this section, we describe how to formulate the optimization problems in Section 3 as mixed-integer programs. We start by presenting a MIP formulation for the optimization problem we solve in the $\mathsf{FindAction}(\boldsymbol{x}, A(\boldsymbol{x}))$ and $\mathsf{IsReachable}(\boldsymbol{x}, \boldsymbol{x}', A(\boldsymbol{x}))$ routines. Next, we describe how this formulation can be extended to the complex actionability constraints in Table 1.

### B.1   MIP FORMULATIONS

Given a point $\boldsymbol{x} \in \mathcal{X}$, an action set $A(\boldsymbol{x})$, and a set of previous optima $\mathcal{A}^{\mathrm{opt}}$, we can formulate $\mathsf{FindAction}(\boldsymbol{x}, A(\boldsymbol{x}))$ as the following mixed-integer program:

$$\min_{\boldsymbol{a}} \quad \sum_{j \in [d]} a_j^+ + a_j^-$$

$$
\begin{array}{llll}
\text{s.t.} & a_j^+ \geq a_j & j \in [d] & \textit{positive component of } a_j & \text{(9a)} \\
& a_j^- \geq -a_j & j \in [d] & \textit{negative component of } a_j & \text{(9b)} \\
& a_j = a_{j,k} + \delta_{j,k}^+ - \delta_{j,k}^- & j \in [d], \boldsymbol{a}_k \in \mathcal{A}^{\mathrm{opt}} & \textit{distance from prior actions} & \text{(9c)} \\
& \varepsilon_{\min} \leq \sum_{j \in [d]} (\delta_{j,k}^+ + \delta_{j,k}^-) & \boldsymbol{a}_k \in \mathcal{A}^{\mathrm{opt}} & \textit{any solution is } \varepsilon_{\min} \textit{ away from } \boldsymbol{a}_k & \text{(9d)} \\
& \delta_{j,k}^+ \leq M_{j,k}^+ u_{j,k} & j \in [d], \boldsymbol{a}_k \in \mathcal{A}^{\mathrm{opt}} & \delta_{j,k}^+ > 0 \implies u_{j,k} = 1 & \text{(9e)} \\
& \delta_{j,k}^- \leq M_{j,k}^- (1 - u_{j,k}) & j \in [d], \boldsymbol{a}_k \in \mathcal{A}^{\mathrm{opt}} & \delta_{j,k}^- > 0 \implies u_{j,k} = 0 & \text{(9f)} \\
& a_j \in A_j(\boldsymbol{x}) & j \in [d] & \textit{separable actionability constraints on } j & \text{(9g)} \\
& a_j^+, a_j^- \in \mathbb{R}_+ & j \in [d] & \textit{absolute value of } a_j & \text{(9h)} \\
& \delta_{j,k}^+, \delta_{j,k}^- \in \mathbb{R}_+ & j \in [d] & \textit{signed distances from } a_{j,k} & \text{(9i)} \\
& u_{j,k} \in \{0,1\} & j \in [d] & u_{j,k} := \mathbb{1}[\delta_{j,k}^+ > 0] & \text{(9j)}
\end{array}
$$

The formulation searches for an action in the set $\boldsymbol{a} \in A(\boldsymbol{x})/\mathcal{A}^{\mathrm{opt}}$ by combining two kinds of constraints: (i) constraints to restrict actions $\boldsymbol{a} \in A(\boldsymbol{x})$ and (ii) constraints to rule out actions in $\boldsymbol{a} \in \mathcal{A}^{\mathrm{opt}}$.

The formulation encodes the separable constraints in $A(\boldsymbol{x})$ – i.e., a constraint that can be enforced for each feature. The formulation must be extended with additional variables and constraints to handle constraints as discussed in Appendix B.2. These constraints are handled through the $a_j \in A_j(\boldsymbol{x})$ conditions in Constraint 9g. This constraint can handle a number of actionability constraints that can be passed solver when defining the variables $a_j$, including *bounds* (e.g., $a_j \in [-x_j, 10 - x_j]$), *integrality* (e.g., $a_j \in \{0,1\}$ or $a_j \in \{L - x_j, L - x_j + 1, \ldots, U - x_j\}$), and *monotonicity* (e.g., $a_j \geq 0$ or $a_j \leq 0$).

The formulation rules out actions in $\boldsymbol{a} \in \mathcal{A}^{\mathrm{opt}}$ through the "no good" constraints in Constraints (9c) to (9f). Here, Constraint (9d) ensures feasible actions from previous solutions by at least $\varepsilon_{\min}$. We set to a sufficiently small number $\varepsilon_{\min} := 10^{-6}$ by default, but use larger values when working with discrete feature sets (e.g., $\varepsilon_{\min} = 1$ for cases where every actionable feature is binary or integer-valued). Constraints (9e) and (9f) ensure that either $\delta_{j,k}^+ > 0$ or $\delta_{j,k}^- > 0$. These are "Big-M constraints" where the Big-M parameters can be set to represent the largest value of signed distances. Given an action $a_j \in [a_j^{\mathrm{LB}}, a_j^{\mathrm{UB}}]$, we can set $M_{j,k}^+ := |a_j^{\mathrm{UB}} - a_{j,k})$ and $M_{j,k}^- := |a_{j,k} - a_j^{\mathrm{LB}}|$.

The formulation chooses each action in $\boldsymbol{a} \in A(\boldsymbol{x})/\mathcal{A}^{\mathrm{opt}}$ to minimize the $L_1$ norm. We compute the $L_1$-norm component-wise as $|a_j| := a_j^+ + a_j^-$ where the variables $a_j^+$ and $a_j^-$ are set to the positive and negative components of $|a_j|$ in Constraints (9a) and (9b). This choice of objective is meant to induce sparsity among the actions we recover by repeatedly solving Algorithm 1.

**MIP Formulation for** $\mathsf{IsReachable}$    Given a point $\boldsymbol{x} \in \mathcal{X}$, an action set $A(\boldsymbol{x})$, we can formulate the optimization problem for $\mathsf{IsReachable}(\boldsymbol{x}, \boldsymbol{x}', A(\boldsymbol{x}))$ as a special case of the MIP in (9) in which we set $\mathcal{A}^{\mathrm{opt}} = \emptyset$ and include the constraint $\boldsymbol{a} = \boldsymbol{x} - \boldsymbol{x}'$. Given that the objective function does not affect the feasibility of the optimization problem, we can set the objective to 1 when solving the problem for $\mathsf{IsReachable}$. In this case, any feasible solution would certify that $\boldsymbol{x}'$ is reachable from $\boldsymbol{x}$ using

the actions in $A(\boldsymbol{x})$. Thus, we can return $\mathsf{IsReachable}(\boldsymbol{x}, \boldsymbol{x}', A(\boldsymbol{x})) = 1$ if the MIP is feasible and $\mathsf{IsReachable}(\boldsymbol{x}, \boldsymbol{x}', A(\boldsymbol{x})) = 0$ if it is infeasible.

## B.2 Encoding Actionability Constraints

We describe how to extend the MIP formulation in (9) to encode salient classes of actionability constraints. Our software includes an ActionSet API that allows practitioners to specify these constraints across each MIP formulation.

**Encoding Preservation for Categorical Attributes**  Many datasets contain subsets of features that reflect the underlying value of a categorical attribute. For example, we may encode a categorical attribute with $K = 3$ categories such marital_status $\in \{single, married, other\}$ using a subset of $K - 1 = 2$ dummy variables such as married and single. In such cases, actions on the dummy variables must obey non-separable actionability constraints to preserve the encoding – i.e., to ensure that a person cannot be married and single at the same time.

We can enforce these conditions by adding the following constraints to the MIP Formulation in (9):

$$L \leq \sum_{j \in \mathcal{J}} x_j + a_j \leq U \tag{10}$$

Here, $\mathcal{J} \subseteq [d]$ is the index set of features with encoding constraints, and $L$ and $U$ are lower and upper limits on the number of features in $\mathcal{J}$ that must hold to preserve an encoding.

Given a standard one-hot encoding of a categorical variable with $K$ categories, $\mathcal{J}$ would contain the indices of $K - 1$ dummy variables for the $K - 1$ categories other than the reference category. We would ensure that all actions preserve this encoding by setting $L = 0$ and $U = 1$.

**Implications and Deterministic Causal Effects**  Datasets often include features where actions on one feature will induce changes in the values and actions for other features. For example, in Table 1, changing is_employed from FALSE to TRUE would change the value of work_hrs_per_week from $0$ to a value $\geq 0$.

We capture these conditions by adding variables and constraints that capture logical implications in action space. In the simplest case, these constraints would relate the values for a pair of features $j, j' \in [d]$ through an if-then condition such as: "if $a_j \geq v_j$ then $a'_j = v_{j'}$". In such cases, we could capture this relationship by adding the following constraints to the MIP Formulation in (9):

$$Mu \geq a_j - v_j + \epsilon \tag{11}$$
$$M(1 - u) \geq v_j - a_j \tag{12}$$
$$uv_{j'} = a_{j'} \tag{13}$$
$$u \in \{0, 1\}$$

The constraints shown above capture the "if-then" condition by introducing a binary variable $u := \mathbb{1}[a_j \geq v_j]$. The indicator is set through the Constraints (11) and (12) where $M := a_j^{\mathsf{UB}} - v_j$ and $\epsilon = 1e - 5$. If the implication is met, then $a_{j'}$ is set to $v_{j'}$ through Constraint (13). We apply this approach to encode a number of salient actionability constraints shown in Table 1 by generalizing the constraint shown above to a setting where: (i) the "if" and "then" conditions to handle subsets of features, and (ii) the implications link actions on mutable features to actions on an immutable feature (i.e. so that actions on a mutable feature years_since_last_application will induce changes in an immutable feature age).

**Generalized Reachability Constraints**  We end with a general-purpose solution to enforce arbitrary actionability constraints on discrete features. These constraints can be used, for example, to preserve a one-hot encoding of ordinal features (e.g., max_degree_BS and max_degree_MS) or a thermometer encoding (e.g., monthly_income_geq_2k, monthly_income_geq_5k, monthly_income_geq_10k).

We can formulate custom reachability constraints for the relevant features $\mathcal{J} \subset [d]$ given two parameters:

1. Set of Viable Values: $V$, a set of all values that can be assigned to the features in $J$.

2. Reachability Matrix: $E \in \{0,1\}^{k \times k}$, a matrix where $e_{i,j} = \mathbb{1}[v_i$ is reachable from $v_j]$ for all $v_i, v_j \in V$.

Given these parameters, we constrain the reachability of features $j \in J$ by adding the following constraints to the MIP formulation in (9):

$$a_j = \sum_{k \in E[i]} e_{i,k} a_{j,k} u_{j,k} \tag{14}$$

$$1 = \sum_{k \in E[i]} u_{j,k} \tag{15}$$

$$u_{j,k} \leq e_{i,k} \tag{16}$$
$$u_{j,k} \in \{0,1\}$$

Here, $u_{j,k} := \mathbb{1}[\boldsymbol{x}' \in V]$ indicates that we choose an action to attain point $\boldsymbol{x}' \in V$. Constraint (14) defines the set of reachable points from $i$, while Constraint (14) ensures that only one such point can be selected. Here, $e_{i,k}$ is $i^{\text{th}}$ row of $E$ for point $i$ and $a_{j,k} := x'_j - x_j$ is the action on feature $j$ to reach point $\boldsymbol{x}' \in V$ from point $\boldsymbol{x}$.

We show an example of how to formulate reachability constraints to preserve a thermometer encoding in Fig. 4.

| | V | | E |
|---|---|---|---|
| NetFractionRevolvingBurdenGeq90 | NetFractionRevolvingBurdenGeq60 | NetFractionRevolvingBurdenLeq30 | |
| 0 | 0 | 0 | $[1,1,0,0]$ |
| 1 | 0 | 0 | $[0,1,0,0]$ |
| 0 | 1 | 0 | $[1,1,1,0]$ |
| 0 | 1 | 1 | $[1,1,1,1]$ |

**Figure 4:** $V$ denotes valid combinations of features. For these features, we wanted to produce actions that would reduce NetFractionRevolvingBurden for consumers. $E$ shows which points can be reached. For example, $[1,1,0,0]$ represents point $[0,0,0]$ can be reached, and point $[1,0,0]$ can be reached, but no other points can be reached.

# C   SUPPLEMENTAL MATERIAL FOR EXPERIMENTS

## C.1   ACTIONABILITY CONSTRAINTS FOR THE german DATASET

We show a list of all features and their separable actionability constraints in Table 3.

| Name | Type | LB | UB | Actionability | Sign |
|------|------|-----|-----|---------------|------|
| Age | $\mathbb{Z}$ | 19 | 75 | No | |
| Male | $\{0,1\}$ | 0 | 1 | No | |
| Single | $\{0,1\}$ | 0 | 1 | No | |
| ForeignWorker | $\{0,1\}$ | 0 | 1 | No | |
| YearsAtResidence | $\mathbb{Z}$ | 0 | 7 | Yes | $+$ |
| LiablePersons | $\mathbb{Z}$ | 1 | 2 | No | |
| Housing=Renter | $\{0,1\}$ | 0 | 1 | No | |
| Housing=Owner | $\{0,1\}$ | 0 | 1 | No | |
| Housing=Free | $\{0,1\}$ | 0 | 1 | No | |
| Job=Unskilled | $\{0,1\}$ | 0 | 1 | No | |
| Job=Skilled | $\{0,1\}$ | 0 | 1 | No | |
| Job=Management | $\{0,1\}$ | 0 | 1 | No | |
| YearsEmployed$\geq$1 | $\{0,1\}$ | 0 | 1 | Yes | $+$ |
| CreditAmt$\geq$1000K | $\{0,1\}$ | 0 | 1 | No | |
| CreditAmt$\geq$2000K | $\{0,1\}$ | 0 | 1 | No | |
| CreditAmt$\geq$5000K | $\{0,1\}$ | 0 | 1 | No | |
| CreditAmt$\geq$10000K | $\{0,1\}$ | 0 | 1 | No | |
| LoanDuration$\leq$6 | $\{0,1\}$ | 0 | 1 | No | |
| LoanDuration$\geq$12 | $\{0,1\}$ | 0 | 1 | No | |
| LoanDuration$\geq$24 | $\{0,1\}$ | 0 | 1 | No | |
| LoanDuration$\geq$36 | $\{0,1\}$ | 0 | 1 | No | |
| LoanRate | $\mathbb{Z}$ | 1 | 4 | No | |
| HasGuarantor | $\{0,1\}$ | 0 | 1 | Yes | $+$ |
| LoanRequiredForBusiness | $\{0,1\}$ | 0 | 1 | No | |
| LoanRequiredForEducation | $\{0,1\}$ | 0 | 1 | No | |
| LoanRequiredForCar | $\{0,1\}$ | 0 | 1 | No | |
| LoanRequiredForHome | $\{0,1\}$ | 0 | 1 | No | |
| NoCreditHistory | $\{0,1\}$ | 0 | 1 | No | |
| HistoryOfLatePayments | $\{0,1\}$ | 0 | 1 | No | |
| HistoryOfDelinquency | $\{0,1\}$ | 0 | 1 | No | |
| HistoryOfBankInstallments | $\{0,1\}$ | 0 | 1 | Yes | $+$ |
| HistoryOfStoreInstallments | $\{0,1\}$ | 0 | 1 | Yes | $+$ |
| CheckingAcct_exists | $\{0,1\}$ | 0 | 1 | Yes | $+$ |
| CheckingAcct$\geq$0 | $\{0,1\}$ | 0 | 1 | Yes | $+$ |
| SavingsAcct_exists | $\{0,1\}$ | 0 | 1 | Yes | $+$ |
| SavingsAcct$\geq$100 | $\{0,1\}$ | 0 | 1 | Yes | $+$ |

**Table 3:** Separable actionability constraints for the german dataset.

The non-separable actionability constraints for this dataset include:

1. DirectionalLinkage: Actions on YearsAtResidence will induce to actions on ['Age']. Each unit change in YearsAtResidence leads to:1.00-unit change in Age

2. DirectionalLinkage: Actions on YearsEmployed$\geq$1 will induce to actions on ['Age']. Each unit change in YearsEmployed$\geq$1 leads to:1.00-unit change in Age

3. ThermometerEncoding: Actions on [CheckingAcctexists, CheckingAcct$\geq$0] must preserve thermometer encoding of CheckingAcct., which can only increase. Actions can only turn on higher-level dummies that are off, where CheckingAcctexists is the lowest-level dummy and CheckingAcct$\geq$0 is the highest-level-dummy.

4. ThermometerEncoding: Actions on [SavingsAcctexists, SavingsAcct$\geq$100] must preserve thermometer encoding of SavingsAcct., which can only increase. Actions can only turn on

higher-level dummies that are off, where `SavingsAcctexists` is the lowest-level dummy and `SavingsAcct≥100` is the highest-level-dummy.

## C.2 ACTIONABILITY CONSTRAINTS FOR THE heloc DATASET

We show a list of all features and their separable actionability constraints in Table 4.

| Name | Type | LB | UB | Actionability | Sign |
|------|------|----|----|---------------|------|
| ExternalRiskEstimate≥40 | $\{0,1\}$ | 0 | 1 | No | |
| ExternalRiskEstimate≥50 | $\{0,1\}$ | 0 | 1 | No | |
| ExternalRiskEstimate≥60 | $\{0,1\}$ | 0 | 1 | No | |
| ExternalRiskEstimate≥70 | $\{0,1\}$ | 0 | 1 | No | |
| ExternalRiskEstimate≥80 | $\{0,1\}$ | 0 | 1 | No | |
| YearsOfAccountHistory | $\mathbb{Z}$ | 0 | 50 | No | |
| AvgYearsInFile≥3 | $\{0,1\}$ | 0 | 1 | Yes | |
| AvgYearsInFile≥5 | $\{0,1\}$ | 0 | 1 | Yes | |
| AvgYearsInFile≥7 | $\{0,1\}$ | 0 | 1 | Yes | |
| MostRecentTradeWithinLastYear | $\{0,1\}$ | 0 | 1 | Yes | |
| MostRecentTradeWithinLast2Years | $\{0,1\}$ | 0 | 1 | Yes | |
| AnyDerogatoryComment | $\{0,1\}$ | 0 | 1 | No | |
| AnyTrade120DaysDelq | $\{0,1\}$ | 0 | 1 | No | |
| AnyTrade90DaysDelq | $\{0,1\}$ | 0 | 1 | No | |
| AnyTrade60DaysDelq | $\{0,1\}$ | 0 | 1 | No | |
| AnyTrade30DaysDelq | $\{0,1\}$ | 0 | 1 | No | |
| NoDelqEver | $\{0,1\}$ | 0 | 1 | No | |
| YearsSinceLastDelqTrade≤1 | $\{0,1\}$ | 0 | 1 | Yes | |
| YearsSinceLastDelqTrade≤3 | $\{0,1\}$ | 0 | 1 | Yes | |
| YearsSinceLastDelqTrade≤5 | $\{0,1\}$ | 0 | 1 | Yes | |
| NumInstallTrades≥2 | $\{0,1\}$ | 0 | 1 | Yes | |
| NumInstallTradesWBalance≥2 | $\{0,1\}$ | 0 | 1 | Yes | |
| NumRevolvingTrades≥2 | $\{0,1\}$ | 0 | 1 | Yes | |
| NumRevolvingTradesWBalance≥2 | $\{0,1\}$ | 0 | 1 | Yes | |
| NumInstallTrades≥3 | $\{0,1\}$ | 0 | 1 | Yes | |
| NumInstallTradesWBalance≥3 | $\{0,1\}$ | 0 | 1 | Yes | |
| NumRevolvingTrades≥3 | $\{0,1\}$ | 0 | 1 | Yes | |
| NumRevolvingTradesWBalance≥3 | $\{0,1\}$ | 0 | 1 | Yes | |
| NumInstallTrades≥5 | $\{0,1\}$ | 0 | 1 | Yes | |
| NumInstallTradesWBalance≥5 | $\{0,1\}$ | 0 | 1 | Yes | |
| NumRevolvingTrades≥5 | $\{0,1\}$ | 0 | 1 | Yes | |
| NumRevolvingTradesWBalance≥5 | $\{0,1\}$ | 0 | 1 | Yes | |
| NumInstallTrades≥7 | $\{0,1\}$ | 0 | 1 | Yes | |
| NumInstallTradesWBalance≥7 | $\{0,1\}$ | 0 | 1 | Yes | |
| NumRevolvingTrades≥7 | $\{0,1\}$ | 0 | 1 | Yes | |
| NumRevolvingTradesWBalance≥7 | $\{0,1\}$ | 0 | 1 | Yes | |
| NetFractionInstallBurden≥10 | $\{0,1\}$ | 0 | 1 | Yes | |
| NetFractionInstallBurden≥20 | $\{0,1\}$ | 0 | 1 | Yes | |
| NetFractionInstallBurden≥50 | $\{0,1\}$ | 0 | 1 | Yes | |
| NetFractionRevolvingBurden≥10 | $\{0,1\}$ | 0 | 1 | Yes | |
| NetFractionRevolvingBurden≥20 | $\{0,1\}$ | 0 | 1 | Yes | |
| NetFractionRevolvingBurden≥50 | $\{0,1\}$ | 0 | 1 | Yes | |
| NumBank2NatlTradesWHighUtilizationGeq2 | $\{0,1\}$ | 0 | 1 | Yes | $+$ |

**Table 4:** Separable actionability constraints for the heloc dataset.

The non-separable actionability constraints for this dataset include:

1. DirectionalLinkage: Actions on `NumRevolvingTradesWBalance≥2` will induce to actions on ['NumRevolvingTrades≥2']. Each unit change in `NumRevolvingTradesWBalance≥2` leads to: 1.00-unit change in `NumRevolvingTrades≥2`

2. DirectionalLinkage: Actions on `NumInstallTradesWBalance≥2` will induce to actions on ['NumInstallTrades≥2']. Each unit change in `NumInstallTradesWBalance≥2` leads to: 1.00-unit change in `NumInstallTrades≥2`

3. DirectionalLinkage: Actions on `NumRevolvingTradesWBalance≥3` will induce to actions on ['NumRevolvingTrades≥3']. Each unit change in `NumRevolvingTradesWBalance≥3` leads to: 1.00-unit change in `NumRevolvingTrades≥3`

4. DirectionalLinkage: Actions on `NumInstallTradesWBalance`≥3 will induce to actions on ['NumInstallTrades≥3']. Each unit change in `NumInstallTradesWBalance`≥3 leads to: 1.00-unit change in `NumInstallTrades`≥3

5. DirectionalLinkage: Actions on `NumRevolvingTradesWBalance`≥5 will induce to actions on ['NumRevolvingTrades≥5']. Each unit change in `NumRevolvingTradesWBalance`≥5 leads to: 1.00-unit change in `NumRevolvingTrades`≥5

6. DirectionalLinkage: Actions on `NumInstallTradesWBalance`≥5 will induce to actions on ['NumInstallTrades≥5']. Each unit change in `NumInstallTradesWBalance`≥5 leads to: 1.00-unit change in `NumInstallTrades`≥5

7. DirectionalLinkage: Actions on `NumRevolvingTradesWBalance`≥7 will induce to actions on ['NumRevolvingTrades≥7']. Each unit change in `NumRevolvingTradesWBalance`≥7 leads to: 1.00-unit change in `NumRevolvingTrades`≥7

8. DirectionalLinkage: Actions on `NumInstallTradesWBalance`≥7 will induce to actions on ['NumInstallTrades≥7']. Each unit change in `NumInstallTradesWBalance`≥7 leads to: 1.00-unit change in `NumInstallTrades`≥7

9. DirectionalLinkage: Actions on `YearsSinceLastDelqTrade`≤1 will induce to actions on ['YearsOfAccountHistory']. Each unit change in `YearsSinceLastDelqTrade`≤1 leads to: -1.00-unit change in `YearsOfAccountHistory`

10. DirectionalLinkage: Actions on `YearsSinceLastDelqTrade`≤3 will induce to actions on ['YearsOfAccountHistory']. Each unit change in `YearsSinceLastDelqTrade`≤3 leads to: -3.00-unit change in `YearsOfAccountHistory`

11. DirectionalLinkage: Actions on `YearsSinceLastDelqTrade`≤5 will induce to actions on ['YearsOfAccountHistory']. Each unit change in `YearsSinceLastDelqTrade`≤5 leads to: -5.00-unit change in `YearsOfAccountHistory`

12. ReachabilityConstraint: The values of [`MostRecentTradeWithinLastYear`, `MostRecentTradeWithinLast2Years`] must belong to one of 4 values with custom reachability conditions.

13. ThermometerEncoding: Actions on [`YearsSinceLastDelqTrade`≤1, `YearsSinceLastDelqTrade`≤3, `YearsSinceLastDelqTrade`≤5] must preserve thermometer encoding of YearsSinceLastDelqTradeleq., which can only decrease. Actions can only turn off higher-level dummies that are on, where `YearsSinceLastDelqTrade`≤1 is the lowest-level dummy and `YearsSinceLastDelqTrade`≤5 is the highest-level-dummy.

14. ThermometerEncoding: Actions on [`AvgYearsInFile`≥3, `AvgYearsInFile`≥5, `AvgYearsInFile`≥7] must preserve thermometer encoding of AvgYearsInFilegeq., which can only increase. Actions can only turn on higher-level dummies that are off, where `AvgYearsInFile`≥3 is the lowest-level dummy and `AvgYearsInFile`≥7 is the highest-level-dummy.

15. ThermometerEncoding: Actions on [`NetFractionRevolvingBurden`≥10, `NetFractionRevolvingBurden`≥20, `NetFractionRevolvingBurden`≥50] must preserve thermometer encoding of NetFractionRevolvingBurdengeq., which can only decrease. Actions can only turn off higher-level dummies that are on, where `NetFractionRevolvingBurden`≥10 is the lowest-level dummy and `NetFractionRevolvingBurden`≥50 is the highest-level-dummy.

16. ThermometerEncoding: Actions on [`NetFractionInstallBurden`≥10, `NetFractionInstallBurden`≥20, `NetFractionInstallBurden`≥50] must preserve thermometer encoding of NetFractionInstallBurdengeq., which can only decrease. Actions can only turn off higher-level dummies that are on, where `NetFractionInstallBurden`≥10 is the lowest-level dummy and `NetFractionInstallBurden`≥50 is the highest-level-dummy.

17. ThermometerEncoding: Actions on [`NumRevolvingTradesWBalance`≥2, `NumRevolvingTradesWBalance`≥3, `NumRevolvingTradesWBalance`≥5, `NumRevolvingTradesWBalance`≥7] must preserve thermometer encoding of NumRevolvingTradesWBalancegeq., which can only decrease. Actions can only turn off higher-level dummies that are on, where `NumRevolvingTradesWBalance`≥2 is the lowest-level dummy and `NumRevolvingTradesWBalance`≥7 is the highest-level-dummy.

18. ThermometerEncoding: Actions on [`NumRevolvingTrades`≥2, `NumRevolvingTrades`≥3, `NumRevolvingTrades`≥5, `NumRevolvingTrades`≥7] must preserve thermometer encoding of

NumRevolvingTradesgeq., which can only decrease. Actions can only turn off higher-level dummies that are on, where `NumRevolvingTrades≥2` is the lowest-level dummy and `NumRevolvingTrades≥7` is the highest-level-dummy.

19. ThermometerEncoding: Actions on [`NumInstallTradesWBalance≥2`, `NumInstallTradesWBalance≥3`, `NumInstallTradesWBalance≥5`, `NumInstallTradesWBalance≥7`] must preserve thermometer encoding of NumInstall-TradesWBalancegeq., which can only decrease. Actions can only turn off higher-level dummies that are on, where `NumInstallTradesWBalance≥2` is the lowest-level dummy and `NumInstallTradesWBalance≥7` is the highest-level-dummy.

20. ThermometerEncoding: Actions on [`NumInstallTrades≥2`, `NumInstallTrades≥3`, `NumInstallTrades≥5`, `NumInstallTrades≥7`] must preserve thermometer encoding of NumInstallTradesgeq., which can only decrease. Actions can only turn off higher-level dummies that are on, where `NumInstallTrades≥2` is the lowest-level dummy and `NumInstallTrades≥7` is the highest-level-dummy.

### C.3 ACTIONABILITY CONSTRAINTS FOR THE `givemecredit` DATASET

We present a list of all features and their separable actionability constraints in Table 5.

| Name | Type | LB | UB | Actionability | Sign |
|---|---|---|---|---|---|
| Age≤24 | $\{0,1\}$ | 0 | 1 | No | |
| Age_bt_25_to_30 | $\{0,1\}$ | 0 | 1 | No | |
| Age_bt_30_to_59 | $\{0,1\}$ | 0 | 1 | No | |
| Age≥60 | $\{0,1\}$ | 0 | 1 | No | |
| NumberOfDependents=0 | $\{0,1\}$ | 0 | 1 | No | |
| NumberOfDependents=1 | $\{0,1\}$ | 0 | 1 | No | |
| NumberOfDependents≥2 | $\{0,1\}$ | 0 | 1 | No | |
| NumberOfDependents≥5 | $\{0,1\}$ | 0 | 1 | No | |
| DebtRatio≥1 | $\{0,1\}$ | 0 | 1 | Yes | + |
| MonthlyIncome≥3K | $\{0,1\}$ | 0 | 1 | Yes | + |
| MonthlyIncome≥5K | $\{0,1\}$ | 0 | 1 | Yes | + |
| MonthlyIncome≥10K | $\{0,1\}$ | 0 | 1 | Yes | + |
| CreditLineUtilization≥10.0 | $\{0,1\}$ | 0 | 1 | Yes | |
| CreditLineUtilization≥20.0 | $\{0,1\}$ | 0 | 1 | Yes | |
| CreditLineUtilization≥50.0 | $\{0,1\}$ | 0 | 1 | Yes | |
| CreditLineUtilization≥70.0 | $\{0,1\}$ | 0 | 1 | Yes | |
| CreditLineUtilization≥100.0 | $\{0,1\}$ | 0 | 1 | Yes | |
| AnyRealEstateLoans | $\{0,1\}$ | 0 | 1 | Yes | + |
| MultipleRealEstateLoans | $\{0,1\}$ | 0 | 1 | Yes | + |
| AnyCreditLinesAndLoans | $\{0,1\}$ | 0 | 1 | Yes | + |
| MultipleCreditLinesAndLoans | $\{0,1\}$ | 0 | 1 | Yes | + |
| HistoryOfLatePayment | $\{0,1\}$ | 0 | 1 | No | |
| HistoryOfDelinquency | $\{0,1\}$ | 0 | 1 | No | |

**Table 5:** Separable actionability constraints for the `heloc` dataset.

The non-separable actionability constraints for this dataset include:

1. ThermometerEncoding: Actions on [`MonthlyIncome≥3K`, `MonthlyIncome≥5K`, `MonthlyIncome≥10K`] must preserve thermometer encoding of MonthlyIncomegeq., which can only increase.Actions can only turn on higher-level dummies that are off, where `MonthlyIncome≥3K` is the lowest-level dummy and `MonthlyIncome≥10K` is the highest-level-dummy.

2. ThermometerEncoding: Actions on [`CreditLineUtilization≥10.0`, `CreditLineUtilization≥20.0`, `CreditLineUtilization≥50.0`, `CreditLineUtilization≥70.0`, `CreditLineUtilization≥100.0`] must preserve thermometer encoding of CreditLineUtilizationgeq., which can only decrease. Actions can only turn

off higher-level dummies that are on, where `CreditLineUtilization≥10.0` is the lowest-level dummy and `CreditLineUtilization≥100.0` is the highest-level-dummy.

3. ThermometerEncoding: Actions on [`AnyRealEstateLoans`, `MultipleRealEstateLoans`] must preserve thermometer encoding of continuousattribute., which can only decrease. Actions can only turn off higher-level dummies that are on, where `AnyRealEstateLoans` is the lowest-level dummy and `MultipleRealEstateLoans` is the highest-level-dummy.

4. ThermometerEncoding: Actions on [`AnyCreditLinesAndLoans`, `MultipleCreditLinesAndLoans`] must preserve thermometer encoding of continuousattribute., which can only decrease. Actions can only turn off higher-level dummies that are on, where `AnyCreditLinesAndLoans` is the lowest-level dummy and `MultipleCreditLinesAndLoans` is the highest-level-dummy.

## C.4 RESULTS ON CLASSIFIER PERFORMANCE

In Table 6, we report the performance of models on all datasets using all algorithms. We split each dataset into a training sample (80%, used for training and hyperparameter tuning) and a hold-out sample (20%, used to evaluate out-of-sample performance).

| Dataset | Model | AUC | | Error | |
|---|---|---|---|---|---|
| | | Train | Test | Train | Test |
| heloc | LR | 0.7723 | 0.7882 | 0.2774 | 0.2774 |
| | XGB | 0.7721 | 0.7880 | 0.2783 | 0.2783 |
| | RF | 0.8593 | 0.7853 | 0.2877 | 0.2877 |
| german | LR | 0.8193 | 0.7602 | 0.2350 | 0.2350 |
| | XGB | 0.8191 | 0.7614 | 0.2300 | 0.2300 |
| | RF | 0.9708 | 0.7937 | 0.2350 | 0.2350 |
| givemecredit | LR | 0.8411 | 0.8441 | 0.2390 | 0.2390 |
| | XGB | 0.8412 | 0.8442 | 0.2380 | 0.2380 |
| | RF | 0.8752 | 0.7928 | 0.2619 | 0.2619 |

**Table 6:** Overview of model performance

## C.5 RESULTS ON THE ILLUSION OF FEASIBILITY

We present the results of an ablation study to show how recourse may appear to be feasible when we fail to consider complex actionability constraints. Here, we repeat the experiments in Section 4 for the `heloc` dataset over three classes of nested actionability constraints:

- Simple, a separable action set which only includes constraints to conditions on the immutability, integrality, and soundness of features.
- Separable, a separable action set which includes all conditions in Simple and adds monotonicity constraints to ensure that certain features can only increase or decrease.
- Actual, a non-separable action set which includes all conditions in Simple and Separable. Note that this corresponds to the action set that we use in our main study.

We present the results from our procedure for all three action sets in Table 7.

| Model Type | Metrics | Actual | | | Separable | | Simple | |
|---|---|---|---|---|---|---|---|---|
| | | Reach | AR | DiCE | AR | DiCE | AR | DiCE |
| LR | Certifies No Recourse | 22.2% | — | — | — | — | — | — |
| | Outputs Action | 77.8% | 85.9% | 57.6% | 85.9% | 57.0% | 99.9% | 65.6% |
| | ↳ Loopholes | 0.0% | 41.1% | 34.4% | 41.1% | 34.8% | 95.5% | 50.3% |
| | Outputs No Action | 22.2% | 14.1% | 42.4% | 14.1% | 43.0% | 0.1% | 34.4% |
| | ↳ Blindspots | 0.0% | 0.0% | 21.0% | 0.0% | 21.7% | 0.0% | 14.6% |
| XGB | Certifies No Recourse | 22.3% | | — | | — | | — |
| | Outputs Action | 77.7% | | 57.3% | | 57.5% | | 60.5% |
| | ↳ Loopholes | 0.0% | NA | 42.1% | NA | 42.0% | NA | 46.7% |
| | Outputs No Action | 22.3% | | 42.7% | | 42.5% | | 39.5% |
| | ↳ Blindspots | 0.0% | | 21.1% | | 21.1% | | 18.2% |
| RF | Certifies No Recourse | 31.3% | | — | | — | | — |
| | Outputs Action | 68.7% | | 49.3% | | 49.3% | | 59.2% |
| | ↳ Loopholes | 0.0% | NA | 29.5% | NA | 29.5% | NA | 44.8% |
| | Outputs No Action | 31.3% | | 50.7% | | 50.7% | | 40.8% |
| | ↳ Blindspots | 0.0% | | 19.8% | | 19.7% | | 15.7% |

**Table 7:** Feasibility of recourse across model classes, and various actionability constraints on the `heloc` dataset. We determine the ground-truth feasibility of recourse using reachable sets (Reach), and use these results to evaluate the reliability of verification with baseline methods for recourse provision (AR and DiCE). We describe the metrics in the caption of Table 2.

