# OpenReview forum: "Prediction without Preclusion: Recourse Verification with Reachable Sets"
_ICLR.cc/2024/Conference — ICLR 2024 spotlight_

### Official Review · Reviewer_2QxH · 2023-10-28

**Soundness:** 2 fair
**Presentation:** 4 excellent
**Contribution:** 2 fair
**Rating:** 5
**Confidence:** 3

**Summary:**

This paper introduces the "recourse verification" task, aimed at identifying models that assign fixed predictions, and proposes methods to assess whether a model can provide actionable recourses based on reachable sets.

**Strengths:**

- The paper is well-written and easy to follow.
- The motivation and justification for fixed points and regions are sound.
- The experiments demonstrate an improvement in the feasibility of recourse across published datasets.

**Weaknesses:**

Weaknesses:
- I am unsure about the paper's significant contribution. The primary focus appears to be on describing a search algorithm to confirm the existence of a feasible action for a user.
- My major concern regarding this paper is its inapplicability to continuous features, as claimed by the authors. Is it possible to extend the MIP formulation to MILP formulation to incorporate continuous features?
- The paper compares the proposed method to two conventional baselines in terms of improving recourse feasibility. Recently, there have been some papers potentially improving feasibility, such as [1] and [2]. I suggest comparing with them.

**Questions:**

- The optimization problem (2) aims to optimize a constant value of 1. What does this objective imply? Does this optimization problem solely seek to find all feasible actions (feasible recourses)?
- Is there a relationship between two reachable sets? For instance, if $x_1$ is within the reachable set of $x$ and $x_2$ is within the reachable set of $x_1$, is it guaranteed that $x_2$ is also within the reachable set of $x$?

**References**:

[1] Nguyen, Duy, Ngoc Bui, and Viet Anh Nguyen. "Feasible Recourse Plan via Diverse Interpolation." International Conference on Artificial Intelligence and Statistics, 2023.

[2] Rafael Poyiadzi, Kacper Sokol, Raul Santos-Rodriguez, Tijl De Bie, and Peter Flach. Face: Feasible and Actionable counterfactual explanations. In Proceedings of the AAAI/ACM Conference on AI, Ethics, and Society, 2020.

---

> ### Author Response · Authors · 2023-11-17
>
> Thank you for your review and questions! We have addressed your questions below
>
> > The paper compares the proposed method to two conventional baselines in terms of improving recourse feasibility.
>
> Thanks for asking this question! This question provides an excellent opportunity to clarify the concept of "feasibility".
>
> These methods are designed to find actions that are “feasible” in that they cross over the decision boundary of complex models like XGB and DNNs. It’s important to note that their primary focus is not on guaranteeing feasibility concerning "actionability constraints."
>
> The recourse problem is:
> \begin{align}
>         \min \quad & cost({\mathbf{a} | \mathbf{x}})\\
>         s.t. \quad & f(\mathbf{x} + \mathbf{a}) = +1 \\
>         & \mathbf{a} \in A({\mathbf{x}})
> \end{align}
>
> The methods you describe are designed to return actions that obey the $f(x+a) = 1$ constraint. However, they cannot handle the actionability constraints $\mathbf{a} \in A(\mathbf{x})$. So their “actions” will violate the $\mathbf{a} \in A(\mathbf{x})$ condition. For example, Face [2] fails to handle interval, encoding, and implication constraints (see Table 1). In practice, this means that they will behave in the same way as DiCE which fails to handle similar constraints. This leads to loopholes (method returns an action that violates the actionability constraints) and blindspots (a point that has a feasible action, but the search algorithm cannot find it) as seen in Table 3
>
> > I am unsure about the paper's significant contribution. The primary focus appears to be on describing a search algorithm to confirm the existence of a feasible action for a user.
>
> This is a very difficult problem to address (please see [this](https://openreview.net/forum?id=SCQfYpdoGE&noteId=AEN6qNww0p) for more details). Stepping back, however, the technical contribution is that it’s hard to build a method that can find actions that will work with general classes of models and that satisfy actionability constraints.
>
> The reason why we need methods like the ones above is that it is inherently difficult to find points that cross over the decision boundary when we work with complex models like DNNs, XGB, and RF. The methods resolve this issue to ensure feasibility with respect to the $f(\mathbf{x}+\mathbf{a})$. In doing so, they sacrifice the ability to certify infeasibility with respect to the $\mathbf{a} \in A(\mathbf{x})$ constraints.
>
> This is where reachable sets are powerful. Our framework provides a way to handle this for any model. We can build a reachable set once and then apply it to many models $f$. This allows us to certify feasibility for complex models like XGBoost, DNNs, and RFs.
>
> > My major concern regarding this paper is its inapplicability to continuous features, as claimed by the authors.
>
> This seems to be a misunderstanding of something that we wrote.
>
> Our approach is applicable to continuous features! Every method that we have in the paper will work on classifiers that use continuous features. Our software is designed to handle them.
>
> The only issue that we have with continuous features is that we may not be able to certify infeasibility in all possible cases. In this regime, we still obtain meaningful results with regard to verification – by certifying feasibility and by flagging certain kinds of infeasibility.
>
> We can show that recourse is feasible by using IsReachable to show that we can reach other points in the sample. Or by certifying that a point produced using another method obeys actionability constraints.
> We can flag infeasibility by detecting only fixed points by calling the FindAction Routine (Eq 5).
> In the remaining, our methods will still work and output results that are correct as we will claim “I don’t know” rather than “feasible” and “infeasible”.
>
> We can show that recourse is feasible by using Theorem 3 with any labeled dataset to set an upper bound for the number of points needed in a reachable set.
>
> > The optimization problem (2) aims to optimize a constant value of 1. What does this objective imply? Does this optimization problem solely seek to find all feasible actions (feasible recourses)
>
> Yes, this is a correct interpretation.
>
> > Is there a relationship between two reachable sets? For instance, if x1 is within the reachable set of x and x2 is within the reachable set of x1, is it guaranteed that x2 is also within the reachable set of x
>
> This is not guaranteed to be true. Suppose $\mathbf{x_1} \in R(\mathbf{x_0})$ and  $\mathbf{x_2} \in R(\mathbf{x_1})$.  $ R(\mathbf{x_0})$ and $R(\mathbf{x_1})$ may or may not be overlapping. In the case when they are not overlapping then $\mathbf{x_2}$ does not lie in $R(\mathbf{x_0})$. In the case when they are overlapping $\mathbf{x_2}$ may lie in $R(\mathbf{x_0})$ if  $\mathbf{x_2}$ is in the overlapping region. The only time when $\mathbf{x_2} \in R(\mathbf{x_0})$ is guaranteed to be in $R(\mathbf{x_0})$ is when $ R(\mathbf{x_1}) \subseteq R(\mathbf{x_0})$.

---

> > ### Comment · Reviewer_2QxH · 2023-11-21
> >
> > Thank you for your response. Some parts of your answer have indeed clarified aspects of your paper for me. I have adjusted the score to 5. However, I would like to discuss a few points further:
> >
> > - Concerning continuous features, I understand that there is uncertainty regarding "infeasibility" answers. The proposal to quantify the probability of "infeasibility" in such cases is both a necessary and a valuable extension.
> >
> > - If I understand correctly, in cases with only discrete features, your problem can be formulated as a graph-based path-finding problem. Each vertex represents a user's feature set, and edges connect vertices with their reachable sets. It seems that Depth-First Search (DFS) or Breadth-First Search (BFS) could be used to determine "feasibility." What distinguishes your method from the approach I've outlined, and what makes your method preferable?
> >
> > - In real-world scenarios, certain actions may be feasible for one group of people but not for others, reflecting the user-dependence feature of reachable sets more broadly. How does your method adapt to such practical cases?

---

> > > ### Author Response · Authors · 2023-11-21
> > > **Author response (1/2)**
> > >
> > > Thank you for increasing your score! We are happy to discuss the points you raised.
> > >
> > > > Concerning continuous features, I understand that there is uncertainty regarding "infeasibility" answers. The proposal to quantify the probability of "infeasibility" in such cases is both a necessary and a valuable extension.
> > > Thank you for this comment!
> > >
> > > There is actually a simple way to quantify the probability of infeasibility in this setting that we can add to the paper. We will describe it below. Before we do, however, note that this is the general-purpose approach for recourse verification when we work with continuous features. We did not include it in this submission because we did not have the space to develop it properly.
> > >
> > > Formally, we can approach these settings by framing recourse verification as a hypothesis test with the form:
> > >
> > >
> > > $H\_0$: At least $\epsilon$ of reachable points from $\mathbf{x}$ have recourse
> > > $H\_1$: $\epsilon$ of reachable points from $\mathbf{x}$ do not have recourse.
> > >
> > > We can then test these hypotheses by generating a sample of $n$ reachable points $S \subseteq R(\mathbf{x})$. Given a classifier, we evaluate its predictions over the points in $S$. If we find that one of the reachable points from $\mathbf{x}$ has recourse, we are done. If we find that none of the reachable points from  $\mathbf{x}$ have recourse, then we can output the following probabilstic guarantee on recourse
> > >
> > >
> > > $Pr(\text{no recourse over }S | H\_0 ) ≤ (1 - \epsilon)^n$
> > >
> > > This is the “p-value” associated with the hypothesis test above and constitutes the formal probabilistic guarantee that you are looking for. For example, suppose we set $\epsilon = 0.01$ and produced a sample $S$ with 1000 reachable points, we would say that:
> > >
> > > Pr(we observe that all points in S have recourse | ≤1% of points have recourse) = 0.99^1000
> > >
> > > Thus we can reject the hypothesis that there exists some small fraction of points with recourse that we didn't observe.
> > >
> > >
> > > The one caveat with this approach is that we will need to generate the samples uniformly. This means that we cannot easily use the reachable sets we obtain. However, we can generate a “random reachable set” by rejection sampling over the subsets of actionable sets. This is simple to implement since we can exploit the same decomposition strategy that we use to generate reachable sets. To be precise, we would decompose the action set over $d$ features into a product of $m$ action sets over subsets of features that can be altered independently. We would then sample a point from each action set using a suitable method. This can be done directly for a single dimensional subspace or via rejection sampling for subspaces of multiple dimensions.
> > >
> > > The only technical challenge associated with algorithm design in this setting arises when we work with action sets where we have a large subset of discrete and continuous features that cannot be altered independently. This situation does not arise that often in practice as it would require an actionability constraint that links together 10 separate features. When it does, however, we would have to treat this as a single block of features and generate a sample from it. Rejection sampling would still work but may require a large number of tries to generate a feasible point.

---

> ### Author Response · Authors · 2023-11-21
> **Author response (2/2)**
>
> > If I understand correctly, in cases with only discrete features, your problem can be formulated as a graph-based path-finding problem. Each vertex represents a user's feature set, and edges connect vertices with their reachable sets.
>
> A graph-based path-finding algorithm would not work for two reasons:
>
> 1. In cases with only discrete features, the problem can only be cast as a graph-based path-finding problem if we have observed all possible samples. Otherwise, we may be interested in looking for paths to vertices that do not exist.
>
> 2. In the case that we had observed all possible features, then reachable points would not necessarily be connected. For example, suppose that the reachable set for the point $\mathbf{x\_0} \in X$ contains the points $\mathbf{x\_1} \in R(\\mathbf{x\_0})$ and $\mathbf{x\_2} \in R(\\mathbf{x\_0})$. In this case, we may have that $\mathbf{x\_2} \not\in R(\mathbf{x\_1})$ and $\mathbf{x\_1} \not\in R(\mathbf{x\_2})$. Please see our previous response for how reachable sets may not be overlapping or contained in one another. Therefore, there may be no path connecting points within the reachable set.
>
>
> > In real-world scenarios, certain actions may be feasible for one group of people but not for others, reflecting the user-dependence feature of reachable sets more broadly. How does your method adapt to such practical cases
>
> Thank you for this question! This would be a setting where - for example, 100 people have the same features $\mathbf{x}\_0$ but different actionability constraints from $A(\mathbf{x}\_0)$. If the people had different features, then they would each have a different reachable set.
>
> In general, the machinery that we develop can be used with any action set at $A(\mathbf{x}_0)$. This means that if we could elicit actionability constraints from each person – using e.g., an interface like [1] - then we could use it to generate a personalized reachable set for each person and use that to verify the feasibility of recourse. In short, we can support this use case. However, we envision that the most common use case is that practitioners will have to run our methods without observing differences among individuals.
>
> In our main use case, we purposefully disregard these individual-level differences and seek to find models that restrict access for all individuals. In this setting, we use an action set that consists of “inherent constraints”. For example, in section 5 we collaborated with a domain expert to identify constraints that would apply to all individuals. $A({\mathbf{x}})$ only includes basic constraints that should adhere to everyone. Thus, when we flag recourse as infeasible using $A(\mathbf{x})$, then it will imply that recourse is also infeasible under $ A\_i({\mathbf{x}})$ since $A({\mathbf{x}}) \subset A\_i({\mathbf{x}}).$ We use this strategy in our experiments specifically for this reason – i.e., to show that recourse can be infeasible simply under actionability constraints that are “not subject to debate.”
>
> [1] Wang, Zijie J, Jennifer Wortman Vaughan, Rich Caruana, and Duen Horng Chau. Gam coach: Towards
> interactive and user-centered algorithmic recourse. In Proceedings of the 2023 CHI Conference on Human
> Factors in Computing Systems, pages 1–20, 2023

---

> ### Comment · Reviewer_2QxH · 2023-11-23
>
> > In cases with only discrete features, the problem can only be cast as a graph-based path-finding problem if we have observed all possible samples. Otherwise, we may be interested in looking for paths to vertices that do not exist.
>
> It's not necessary to construct the entire graph of all samples; the graph-based implication is just a more intuitive concept. Starting from the initial vertex $x_0$, you iterate through the child nodes that are both valid and within the Reachable set $R(x_0)$. For instance, let's consider the child nodes $x_1$ and $x_2$. If either $x_1$ or $x_2$ represents a recourse, the procedure concludes. If not, the procedure continues with the valid child nodes of $x_1$ and then proceeds to $x_2$. Note that in some cases, you do not need to consider all child nodes, just ones that are confirmed valid.
>
> > In the case that we had observed all possible features, then reachable points would not necessarily be connected. For example, suppose that the reachable set for the p ...
>
> The graph I am referring to is specifically a directed graph. I'm having difficulty understanding your message.

---

### Official Review · Reviewer_jEod · 2023-10-31

**Soundness:** 2 fair
**Presentation:** 3 good
**Contribution:** 3 good
**Rating:** 8
**Confidence:** 3

**Summary:**

This paper studies recourse verification of machine learning models. Recourse verification is an important aspect of algorithmic recourse, which seeks to identify models that assign predictions without any actionable recourse for the decision subject. Ensuring the existence of actionable recourse is essential in applications affecting people’s lives and livelihoods, such as job hiring, loan approvals, and welfare programs. A model that offers no recourse for its decisions may permanently exclude subjects from accessing these benefits without offering a path to eligibility. Existing research largely focuses on recourse provision — providing individuals with actionable recourse — but only a few works study the infeasibility of providing recourse.

This work proposes an approach for recourse verification under actionability constraints based on reachable sets. A reachable set is a collection of feature vectors that can be reached from a given input using a set of allowed actions. The proposed method certifies the existence or non-existence of recourses by querying the model on every point in the reachable set or an approximation of this set. If the method finds a subset of the reachable set that contains a recourse, it certifies the existence of recourse. Similarly, if it cannot find a recourse in a superset of the reachable set, it certifies the infeasibility of providing recourse. If it cannot certify either of the above, it abstains.

**Strengths:**

1. The paper studies an important problem that has not been explored well in the literature. It makes a significant contribution in this area.
2. The paper is well-written and easy to follow.
3. It is claimed that the proposed method does not require any assumption on the prediction model. However, the model might need to satisfy some conditions for the decomposition approach, which is essential when the problem dimensionality is high. See the weaknesses section for more details.

**Weaknesses:**

1. The recourse verification process evaluates every point in the reachable set, which could be time-consuming if the problem dimensionality is high. The paper seeks to address this issue by a decomposition approach that partitions the action set using features that can be altered independently. However, this approach has not been explained well in the paper.
2. It is unclear how the separable features are identified. What role does the prediction model play in the identification of these features?
3. It is unclear what conditions the prediction model must satisfy for the features to be separable. For instance, the verification step may return an infeasibility certificate in partitions A_1(x) and A_2(x), but actionable recourses may still exist in the Cartesian product A_1(x) X A_2(x) of the two sets.

Minor comments:
1. Increasing the font size in Tables 1 and 2 could help improve readability.
2. It seems like a word is missing in the following sentences:
    1.  Pg. 1 — “In fraud detection and content moderation, for example, models should assign fixed [predictions?] to prevent malicious actors from…”
    2. Pg. 3 — “We can elicit these constraints from users in natural language and convert them to expressions that can [be] embedded into an optimization problem.”
3. Figure 3 is a bit confusing and could be made clearer. The x-axis has no label. It seems that the size of the reachable set *grows* rapidly under the decomposition approach compared to brute force, which is contrary to the text. If I understand correctly, the purpose of using decomposition is to reduce the number of points to verify.

**Questions:**

1. Could this approach be extended to certify the existence or non-existence of an abundance of recourse options instead of just one? A single recourse option might not be feasible for everybody, and having multiple recourses could provide more options to people. It might be possible to certify statements like "20% of the actions in the action set would lead to a positive outcome" by querying a random subset of the action set.
2. Different actions may have different costs for the subjects. For instance, it might be easier for a loan applicant to increase their credit score than their income. Could we incorporate costs for the actions and certify the existence of a low-cost recourse?

---

> ### Author Response · Authors · 2023-11-17
>
> Thank you for your time and feedback. We appreciate your suggestions for minor comments and will address them! We would like to help clarify the misunderstandings.
>
> > It is unclear how the separable features are identified. What role does the prediction model play in the identification of these feature?
>
> There seems to be some kind of misunderstanding that we would like to correct. The prediction model plays no role in determining if features are separable or non-separable.
> Separable features are defined as features that can be altered independently under a given action set $A(x)$. For example, the features $reapplicant$ and $n\\_accounts$ can be altered independently of one another. However, features like $max\\_degree\\_BS$, $max\\_degree\\_MS$, $max\\_degree\\_PHD$ that arise from a one-hot encoding cannot be altered independently. Likewise, features that are tied together in $X$ like if $is\\_employed = true$ then $work\\_hrs\\_per\\_week > 0$.
> In our case, the action set captures all the relationships described above. Given an action set $A(\mathbf{x})$, we can easily identify the features that can be altered independently if the actions from one feature $A(x_1)$ do not affect or are not dependent on the actions in $A(x_2)$. Likewise we can find the actions that are non-separable if the actions in $A(x_1)$ rely on other features.  We can express the action set as a  “product space” of smaller action sets over the features. We can use this to _generate_ reachable sets for each action set in a way that is far more efficient without sacrificing correctness.
>
> > the verification step may return infeasibility certificate in partitions A_1(x) and A_2(x), but actionable recourses may still exist in the Cartesian product A_1(x) X A_2(x) of the two sets.
>
> This seems to be a misunderstanding that stems from the previous comment on separability. This cannot happen because our partition is designed to specifically avoid this effect. In particular, we would only consider a partition where we can express the actions as $A(x) = A\_1(x_1) \times A\_2(x_2) \times A_m(x_m)$
>
> > Could this approach be extended to certify the existence or non-existence of an abundance of recourse options instead of just one.
> A single recourse option might not be feasible for everybody.
>
> We thank you for bringing this to our attention!
>
> Our methods can easily be extended to certify the existence or non-existence of multiple options. We can count the number of recourse options in a reachable set as a measure of “abundance.” Given a complete reachable set, we can certify existence or non-existence. Given an interior approximation, we can only certify existence.
> Stepping back, we think that the measures are also interesting as a richer measure of feasibility when recourse is feasible - e.g., if there is only 1 point in a reachable set that would lead to recourse, then recourse is brittle and existing methods for recourse provision are unlikely to find it (i.e., high likelihood of blind spot). We will mention this in our manuscript!
>
> > Could we incorporate costs for the actions and certify the existence of a low-cost recourse?
>
> Yes this can be done. Here are the steps:
> Compute the reachable set  such that $\\{ \mathbf{x} + \mathbf{a} \mid \mathbf{a} \in A({\mathbf{x})} \text{ and } \mathbf{a} \in \text{cost}(\mathbf{a};\mathbf{x}) \\}$
> We can now filter the reachable set $R(\mathbf{x})$ to only include reachable points such that $\text{cost}(\mathbf{x},\mathbf{x_0}) ≤ B$ where $B$ represents a budget. Filtering out points after the reachable set has been computed allows us to generate the reachable set once and reuse it to handle different budgets.
> We do not explore this for two reasons. The first is because costs and budgets are difficult to elicit in a way that most people would agree on. The second, relatedly, is because we would like to flag infeasibility in a way that is not subject to debate. So if a model is precluded access, this is arising because of the oversight of basic actionability constraints rather than because we have elicited the wrong costs or set the wrong budget.
>
> > the purpose of using decomposition is to reduce the number of points to verify
>
> We want to clarify that the purpose of decomposition is not “reduce the number of points to verify.” Rather it is a computationally efficient way to compute the reachable set $R(\mathbf{x})$. The action set remains the same regardless of decomposition. Figure 3 represents just how efficient decomposition is compared to brute force. In less than 3 seconds decomposition is able to fully compute large reachable sets of 600 points compared to brute force which can only generate 40 points in the same time frame.
>
> > Minor comments
>
> Thank you for catching these. We have uploaded an updated manuscript with these fixes!

---

> > ### Comment · Reviewer_jEod · 2023-11-23
> > **Reviewer Comment**
> >
> > I would like to thank the authors for their time and effort in responding to my review. The authors have addressed my concerns and explained the parts that were unclear to me. Including those clarifications in the paper might help make it easier to understand.
> >
> > In my opinion, this paper studies an important problem and makes a significant contribution towards solving it, which justifies its acceptance. I am happy to revise my recommendation to accept.

---

### Official Review · Reviewer_z4uU · 2023-11-01

**Soundness:** 2 fair
**Presentation:** 2 fair
**Contribution:** 2 fair
**Rating:** 5
**Confidence:** 2

**Summary:**

This work introduces the recourse verification: to verify if the prediction is desirable for any actions over the inputs, which is modeled as a formal verification problem given the trained model and input specifications. The paper gives examples using the proposed reachable sets.

**Strengths:**

- The idea of verification seems to be novel in the sense of recourse and the motivation is clear.
- The formulation is easy to follow.

**Weaknesses:**

- My biggest concern lies in the lack of contribution in the verification methods, which directly follow the basic idea of formal verification but seem not to dive deeper into the optimization algorithms or target the specific challenge in the recourse setting.
- When introducing reachable sets, more details are expected to be discussed, i.e. continuous or discrete, $\ell_p$-norm bound ball. The verification seems to be sound but incomplete, and it is expected to be compared to more off-the-shelf reachibility-based verification methods in [1].
- Although experiments show the prediction without recourse and current methods fail to detect them, there are no other baselines of recourse verification for the comparison of tightness and time efficiency. Also, the experiment part is not well organized in the sense of merging section 4 and 5 as experiments.

[1] Liu, C., Arnon, T., Lazarus, C., Strong, C., Barrett, C., & Kochenderfer, M. J. (2021). Algorithms for verifying deep neural networks. *Foundations and Trends® in Optimization*, *4*(3-4), 244-404.

**Questions:**

See weakness

---

> ### Author Response · Authors · 2023-11-17
>
> Thank you for your review and feedback! We found a few big misunderstandings in your review that we would like to address!
>
> > When introducing reachable sets, more details are expected to be discussed, i.e., continuous or discrete, ℓp-norm bound ball. The verification seems to be sound but incomplete, and it is expected to be compared to more off-the-shelf reachability-based verification methods in [1].
>
> There seems to be a big misunderstanding that we would like to clarify! We are very familiar with the reachability-based methods to verify deep neural networks presented by Liu et al. [1]. One issue is that these algorithms are designed for deep neural networks, so they would not naturally work for complex model classes like RFs and XGB. **The main issue is that these algorithms will fail for “recourse verification” because they cannot verify the stability of predictions under *actionable perturbations.***
>
> In particular, the algorithms are designed to certify that the prediction of a DNN at $\mathbf{x}_0$ will not change under perturbations in a smooth and convex set  – e.g., the set of all perturbations within a $\ell_p$-Ball. In a recourse verification task, however, we are exclusively interested in certifying that a model’s predictions at $\mathbf{x}_0$ would not change under the set of actionable perturbations: $\\{ \mathbf{x} = \mathbf{x}_0 + \mathbf{a} \mid \mathbf{a} \in A(\mathbf{x}_0) \\}$
>
> This set of actionable perturbations is neither smooth nor convex. This is because the action set $A(\mathbf{x}_0)$ is neither smooth nor convex in the vast majority of applications we care about (i.e., classification tasks with semantic feature spaces). For example, $A(\mathbf{x})$ would require discrete perturbations for binary and integer-valued features like $n\\_accounts$. This is also the case for non-separable constraints that we need to enforce actionability over categorical variables like job_type or capture logical relationships like if $is\\_employed = \texttt{TRUE} \quad \implies\quad hrs\\_worked\\_per\\_week > 0$.
>
> We are happy to include a comparison in a potential revision. However, we do not think that this would be informative since it would show that the methods fail at a task that they need to be designed to handle. In particular, we would expect to obtain a number of loopholes and blindspots using these approaches as we would have to work with a convex relaxation of the perturbation set.
>
> > There are no other baselines of recourse verification for the comparison of tightness and time efficiency.
>
> This might also be a misunderstanding related to the issue above. This is the first paper to present an actual procedure for recourse verification, so there are no other baselines to compare to. There are also no other off-shelf methods that are designed for recourse verification in this setting – in part because we are exclusively interested in certifying stability over “actionable” perturbations where the perturbation set is almost always non-smooth and non-convex.
>
> With this in mind, we did want to provide a detailed account of tightness and time efficiency. We discuss these issues in Section 3. To reiterate, the MIP is solved to optimality within ≪ 1s for the instances we consider in Section 4. In general, the core property that we exploit is that action sets can be decomposed to compute reachable sets. In Figure 3, for example, we can generate the largest reachable set that we would need in $\leq3$ seconds. Computation and time efficiency are important considerations since they may often present a barrier to adoption. Our results above suggest that these issues should not be a bottleneck in most practical applications. In the event that they are the algorithms have the benefit that they use limited memory and can be easily run in parallel over multiple CPUs. Another key benefit is that we only need to call these procedures once – as the reachable set that we recover can be used for all possible models
>
> > My biggest concern lies in the lack of contribution in the verification methods, which directly follow the basic idea of formal verification
>
> Thank you for mentioning this! Some of this may stem from a lack of familiarity with work on algorithmic recourse. If so, please read our main response above. In short, formal verification is urgently required in this task – since models can assign fixed predictions that permanently preclude access from credit, employment, or assistance. The task is also challenging from a technical standpoint – as we must certify the stability predictions of a model over the set of actionable perturbations, which are neither convex nor smooth. This lack of structure makes it a very difficult problem to address, especially in a way that is model-agnostic. In contrast, existing robustness verification methods (e.g. [1]) primarily focus on structured and nice perturbation sets. We discuss these issues in Section 3, but we are happy to make it more prominent if you need.

---

> > ### Comment · Reviewer_z4uU · 2023-11-23
> >
> > Thanks for the clarification. Some of my concerns have been addressed, and so I will raise my score to 5.

---

### Official Review · Reviewer_fbZh · 2023-11-01

**Soundness:** 2 fair
**Presentation:** 2 fair
**Contribution:** 2 fair
**Rating:** 6
**Confidence:** 3

**Summary:**

The authors present a new idea, recourse verification, certifying if a predictive model guarantees actionable items for users to change the prediction outcome. Different from the typical algorithmic recourse problem where the goal is to find actionable items with minimum cost, this work aims at ensuring that users are not mistakenly precluded from recourse. In the paper, the authors first establish fundamental concepts and theorems for this new topic. Afterwards, they propose "reachable set" for enumerating plausible feature values after actions. With proper decomposition of feature space as the author propose, feasibility of recourse can be effectively tested. Finally, the authors conduct evaluations on real-world datasets and confirm the efficacy of verification.

**Strengths:**

1. Recourse verification as a new research topic seems intriguing and impactful. It makes sense that some predictive models can accidentally limit availability of recourse and thereby hinder the fairness. Upon this important issue, the authors establish a good foundation for follow-up research and may also benefit researchers working on the typical algorithmic recourse problems.
2. The proposed algorithms seem reasonable and the step of implementation is clear. Also, the effectiveness is verified in the experiments.
3. The writing is overall clear and easy to follow. The details of experiments are provided. The limitations of this work are also adequately discussed.

**Weaknesses:**

Certain parts of the proposed method may still be in early stages of development, which may require further refinement to guarantee its practical value. For example, as discussed in the limitation section, the verification algorithm does not work on continuous features. More concerns of mine are summarized in the Questions section below.

**Questions:**

1. It is unclear how often does the undesired preclusion occur in practice. In particular, continuous features are quite common and may trivially avoid preclusion if the capacity of the predictive model is not constrained. Even if we focus on discrete features only, I am still not sure if undesired preclusion can frequently happen. Let us assume users A and B who pass and got rejected respectively by a predictive model. If we ignore the cost, an easy recourse for A can be the difference between A and B in the feature space. If there are more users getting approved by the model, more candidates of recourse are available for A's actions; namely, it is more unlikely that we find no proper recourse for user A when data size grows. If the diversity of the approved users is so limited that no recourse can be found for user A, I wonder if the preclusion is then more like intended (e.g., setting up strict rules) instead of being an accident.
2. Following question 1, I am wondering if it is reasonable to adopt the idea of cost constraint in recourse provision to reduce the reachable set? For example, we certify if a model is not "fixed" given an upper bound of cost.
3. How do we check the quality of a recourse verification algorithm? Specifically, if we employ two recourse verification methods and get inconsistent results, how do we decide which one is better?

---

> ### Author Response · Authors · 2023-11-17
>
> > It is unclear how often does the undesired preclusion occur in practice. In particular, continuous features are quite common and may trivially avoid preclusion if the capacity of the predictive model is not constrained…. If the diversity of the approved users is so limited that no recourse can be found for user A
>
> We think there may be a big misunderstanding, and we’d like to get to the bottom of it. To make things easier, let’s consider an extreme version of your setting. Say we have a classifier $f$ that approves *one* user with features $\mathbf{x}_\textrm{app}$ and rejects all other users with features $\mathbf{x}_i \neq \mathbf{x}\_{app}$ . Formally, we have:
>
> \begin{align}
> f(x) = 1 \text{ for all } \mathbf{x} = \mathbf{x}\_\textrm{app} \\\\
> f(x) = -1 \text{ for all } \mathbf{x} \neq \mathbf{x}\_\textrm{app}
> \end{align}
>
> Given a person with features $\mathbf{x}\_i$, let us define the change they need to reach features $\mathbf{x}\_\textrm{app}$ as $\mathbf{a}\_i = \mathbf{x}\_\textrm{app} - \mathbf{x}\_i$. Your review seems to suggest that we would claim that person $i$ does not have recourse when $\mathbf{a}\_i  \notin A(\mathbf{x}\_i).$ To be clear, this is not the case. When $\mathbf{a}\_i \notin A(\mathbf{x}\_i)$, this only means that person $i$ cannot reach $\mathbf{x}_\textrm{app}$. However, we would not claim “preclusion” because there may be many other points that person $i$ could reach that lead to approval.
>
> This is because we search over all possible changes that a user can perform - not simply the changes to reach users that are approved.
>
> Our methods are designed to handle this search because we search for all possible actions that a person could perform over the feature space (i.e., rather than only the features of users in, e.g., a training dataset). **Thus, the “diversity of users” in a dataset will not influence a claim of preclusion.**
> When the feature set is discrete, we can search over all points in the discrete exhaustively, and thus claim that a person $i$ has recourse or does not have recourse.
> When the feature set is continuous, we can only claim that a person has recourse. If we cannot find a point for which a person has recourse, then we claim that we do not know.
>
> > Following question 1, I am wondering if it is reasonable to adopt the idea of cost constraint in recourse provision to reduce the reachable set? For example, we certify if a model is not "fixed" given an upper bound of cost.
>
> Yes this can be done. Here are the steps:
> Compute the reachable set  such that $\\{ \mathbf{x} + \mathbf{a} \mid \mathbf{a} \in A({\mathbf{x})} \text{ and } \mathbf{a} \in \text{cost}(\mathbf{a};\mathbf{x}) \\}$
> We can now filter the reachable set $R(\mathbf{x})$ to only include reachable points such that $\text{cost}(\mathbf{x},\mathbf{x_0}) ≤ B$ where $B$ represents a budget. Filtering out points after the reachable set has been computed allows us to generate the reachable set once and reuse it to handle different budgets.
>
>
> We do not explore this for two reasons. The first is because costs and budgets are difficult to elicit in a way that most people would agree on. The second, relatedly, is because we would like to flag infeasibility in a way that is not subject to debate. So if a model is precluded access, this is arising because of the oversight of basic actionability constraints rather than because we have elicited the wrong costs or set the wrong budget.
>
>
> > How do we check the quality of a recourse verification algorithm? Specifically, if we employ two recourse verification methods and get inconsistent results, how do we decide which one is better?
>
> We should not get “inconsistent” results. Say that we have two different methods for recourse verification $Recourse_A({\mathbf{x}},{f},{R})$ and $Recourse_B({\mathbf{x}},{f},{R})$
>
> Here, each method $Recourse_A$ and $Recourse_B$ takes as input a model $f$, a point $x$ and a reachable set $R(x)$. Given these methods, it would output a claim of “has recourse”, “no recourse”, or “I don’t know.”
>
> In this case, the results should be consistent in that we should never have one method claim “recourse” and another method “no recourse.”
>
> We can have a situation where $Recourse_A$ or $Recourse_B$ claims “I don’t know” while another claims “recourse” or “no recourse”:
>
> Based on this, methods for recourse verification can only differ in terms of their abstention rate. Ideally, we would like for methods to avoid abstaining. In applications like credit scoring, we would like to have a low “abstention rate” on the points that do not have recourse. This is because we would like to flag models that can detect preclusion.

---

> ### Comment · Reviewer_fbZh · 2023-11-22
>
> I thank the authors for the thorough response. The response addresses my concerns, particularly regarding Question 1. After evaluating this work again, we decide to keep my score and vote for accept. As previously commented, this work has some limitations (e.g. issues on continuous features) as acknowledged by the authors, while the idea of verification seems promising and potentially impactful. I also think the authors have established a solid foundation for this new topic, which can benefit researchers engaged in this area.

---

### Author Response · Authors · 2023-11-17
**Author Rebuttal by Authors**

We thank the reviewers for their time and feedback! We are glad that the reviews remarked that we make “a significant contribution in this area (jEod)“ that it is “important (jEod)”,  “intriguing and impactful” (fbZh), “novel” (fbZh), and “has not been explored well in the literature (jEod)”. They highlight that the paper is “well-written” (jEod, 2QxH) and “clear” (fbZh).

We found a few important misunderstandings in our reviews that we have addressed in our responses to reviews. One overarching concern on our end is that some of our reviews may have missed the urgent need for research that deals with *infeasibility* in the recourse literature. This is an issue that is critical to the originality and impact of our work. Given this, we'd like to provide some additional background and context below.

To put things in perspective, there have been hundreds of papers on recourse since 2018. **The vast majority of these works ignore the fact that recourse may be infeasible.** Only 4 papers state that recourse may be infeasible. Only one suggests that it may arise in practice. As we show, existing methods provide recourse output actions that violate simple constraints on actionability. This has far-reaching consequences because it is a *silent failure.* In applications such as lending, we overlook models that would permanently bar access but present consumers with “reasons without recourse.”

Our work is the first to study the infeasibility of recourse – we not only show that recourse may be infeasible under actionability constraints, but articulate how certifying infeasibility can be used as a standalone procedure to ensure access. Our work highlights a major issue that leads to silent failures in a large class of methods. Moreover, it introduces practical algorithms that can reliably detect this issue in a large class of applications. We see this as a first step for recourse verification, and note that **one of its major contributions is to draw attention to a key aspect of algorithmic recourse that is largely unexplored.**

We hope that this provides some additional information to evaluate the originality and significance of this work, and we look forward to addressing any remaining questions and concerns over the coming days

---

### Meta-Review · Area_Chair_rTDP · 2023-12-10

**Metareview:**

The reviewers were split about this paper and did not come to a consensus: on one hand they appreciated the introduction of the concept of recourse verification, the clarity of the writing, and importance of the problem addressed; on the other they had doubts about (1) how rare preclusion actually is, (2) lack of contribution towards verification methods, (3) several unclear points. After going through the paper and the discussion I have decided to vote to accept because the authors respond convincingly to each of the main concerns. Specifically for (1) the reviewers argued that continuous features may trivially avoid preclusion (i.e., that a person does not have recourse) without constraints on model capacity. Further they were worried that the preclusion depended on who exactly is in the training dataset The authors responded with a clarifying example pointing out that for continuous features they are not able to claim preclusion as they cannot exhaustively search over all points in space. However, for discrete features they can indeed make a claim of preclusion. Further, these claims are not dependent on the training dataset as all points in discrete space are searched. Finally the authors sketch a way to quantify the probability of infeasibility for continuous features. This fully resolves the reviewer concern. For (2), the reviewers argued that the paper did not contribute significantly towards formal verification methods. The authors responded by pointing out that their work is the first to proopse such verification and that it is technically non-trivial as actionable perturbations are neither convex nor smooth, resolving the point. For (3), the reviewers were unclear how separable features were identified and the conditions which the prediction model must satisfy for the features to be separable. However, the authors have convinced me that they are able to fix these uncertainties in the final version in their responses to reviewers. For these reasons I argue for acceptance. Authors: the reviewers have given extremely detailed feedback and I recommend the authors follow / respond to their comments closely, as you have already started to do. Once this is done, the paper will make a great contribution to the conference!

**Justification For Why Not Higher Score:**

The work could have been presented a bit more clearly. The authors did a good job clarifying things in the rebuttal but I'm not certain they could do so in an oral presentation.

**Justification For Why Not Lower Score:**

Paper introduces a verification technique for algorithmic recourse. This is something that needs to be highlighted to the ML community.

---

### Decision · Program_Chairs · 2024-01-16

Accept (spotlight)